# REPETITION IMPROVES LANGUAGE MODEL EMBEDDINGS

**Jacob Mitchell Springer    Suhas Kotha    Daniel Fried    Graham Neubig**
**Aditi Raghunathan**
Carnegie Mellon University

## ABSTRACT

Bidirectional models are considered essential for strong text embeddings. Recent approaches to adapt autoregressive language models (LMs) into strong text embedding models have largely had the requirement to modify the LM architecture to be bidirectional. We challenge this premise by introducing "echo embeddings" which converts autoregressive LMs into high quality text embedding models *without* changing the architecture or requiring fine-tuning. By repeating the input and extracting embeddings from the repeated tokens—which have access to all original tokens—echo embeddings improve over classical LM embeddings by over 5% in zero-shot settings. Our zero-shot embeddings nearly match those obtained by bidirectionally-converted LMs that undergo additional masked-language modeling training. Echo embeddings are also compatible with supervised fine-tuning, matching or outperforming bidirectionally-converted LMs in an apples-to-apples comparison, even with an identical compute budget during training and inference. Overall, repetition is a simple and effective strategy to circumvent the need for bidirectional attention in embedding models, paving the way towards a unified architecture for all NLP tasks.

## 1  INTRODUCTION

Neural text embeddings have a crucial role in modern approaches to information retrieval (IR), semantic similarity estimation, classification, and clustering (Ni et al., 2021b; Muennighoff et al., 2022). For example, document retrieval often leverages low-dimensional embeddings for efficient lookup: when queries and documents are encoded as vectors where semantic relationships are described by similarity in some metric space, a query lookup can be reduced to an approximate nearest-neighbor search in embedding space (Johnson et al., 2019; Vanderkam et al., 2013).

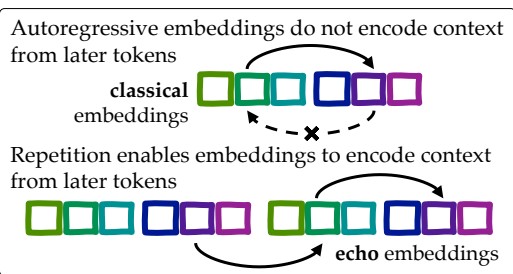

Figure 1: Overview of echo embeddings.

Until recently, the dominant pretrained language model paradigm for neural embeddings have been masked language models with bidirectional attention (Ni et al., 2021a; Raffel et al., 2020; Izacard et al., 2021; Wang et al., 2022; Jiang et al., 2022; Su et al., 2022; Xiao et al., 2023a; Li et al., 2023). Very recent literature has made progress generating high quality embeddings from autoregressive language models. However, these approaches either require large-scale synthetic data (Wang et al., 2023) or for the architecture of the base model to be "cast" to have bidirectional attention (BehnamGhader et al., 2024; Muennighoff et al., 2024). The requirement to change the architecture prevents the extraction of high quality embeddings without additional training. Further, this requirement is an obstacle to the development of truly unified models that can generate text and produce embeddings simultaneously without changing the architecture between encoding text for embeddings and generating tokens.

Is bidirectional attention crucial for high quality embeddings? Intuitively, autoregressive models fail to capture information across the entire input due to the causal attention mask. We show this concretely in a toy setup in Section 3. In this work, we present a surprisingly effective way to overcome this limitation *without* requiring bidirectional attention or fine-tuning.

Our method, "echo embeddings" is simple: we prompt the model with the input *twice* and extract embeddings from the second occurrence of the input. Even with causal attention, the embeddings of the second occurrence can access the entire input in the first occurrence. The second key ingredient is to prompt the language model to repeat the input (or, for example, to fix, rephrase, or otherwise reconstruct the input). This simple change enables the model to extract bidirectional information. As a result, echo embeddings offer strong performance in the entirely zero-shot setting—without any additional fine-tuning.

In comparison with the classical method of extracting embeddings, the zero-shot performance of echo embeddings is around 5% higher. This nearly matches the performance of the recent more expensive method of casting to bidirectional attention followed with an addititional unsupervised MLM training (BehnamGhader et al., 2024). Even in comparison to the strong zero-shot baseline of PromptEOL (Jiang et al., 2023b)—a method in which the model is prompted to summarize the input in a single token—echo embeddings outperform by a significant margin, and is less sensitive to the exact choice of prompt. We emphasize that echo embeddings are simple, cheap, and flexible.

The current best embeddings still require additional fine-tuning with a supervised objective. We find that echo embeddings slightly outperform the classical approach to extracting embeddings, even when the architecture is cast to be bidirectional, and even when echo embeddings is given an identical compute budget. We speculate that these gains are because echo embeddings better preserve and leverage large-scale pretraining because there are no architectural changes. This is in contrast to the bidirectional relaxation approach, which requires the model to adjust its weights to perform well with bidirectional attention.

Echo embeddings come with a lot of benefits: simple to apply, works zero-shot without additional training, allows a unified model to be used for various tasks, and offers (slightly) better performance than casting to bidirectional attention offers. One apparent drawback is that repetition doubles the compute cost. To account for this, in all our experiments, we report "compute matched" results for both inference time and training time (when applicable). While the gains are now slightly smaller, we see that echo embeddings still perform well. In total, we believe that the echo embedding approach, by virtue of its simplicity and effectiveness, is an important step in bridging the gap between traditionally bidirectional embedding models and the current predominant paradigm of autoregressive language models.

## 2 Preliminaries

Our goal is to extract text embeddings that map a sentence $x$ to a vector $\phi(x) \in \mathbb{R}^d$ such that the semantic similarity between sentences is captured as similarity between their embeddings. In practice, we use the cosine similarity between embeddings as a similarity metric (detailed in Appendix C).

**Embeddings from language models.** We are primarily interested in the embeddings extracted from *autoregressive* language models, which typically have causal attention masking and are trained on a next-token objective. For brevity, we drop the term "autoregressive" in the following.

As is standard, we extract embeddings from the activations of the final hidden layer. Each input token $x_j$ at position $j$ is associated with a *contextualized token embedding* which is the hidden layer representation $\phi_j(x)$. We can pool the embeddings across all the tokens in different ways. In this work, we focus on two common strategies which have been considered by prior work (Reimers & Gurevych, 2019; Muennighoff, 2022; Zhang et al., 2023a; Wang et al., 2023).

**Pooling.** A *mean token embedding* over a set of indices $A$, refers to the mean contextualized token embeddings at indices in $A$ : $\phi_A(x) := \frac{1}{|A|} \sum_{t \in A} \phi_t(x)$. A *last-token embedding* refers to the contextualized token embedding of the last token in the input sequence, written $\phi_{-1}(x)$.

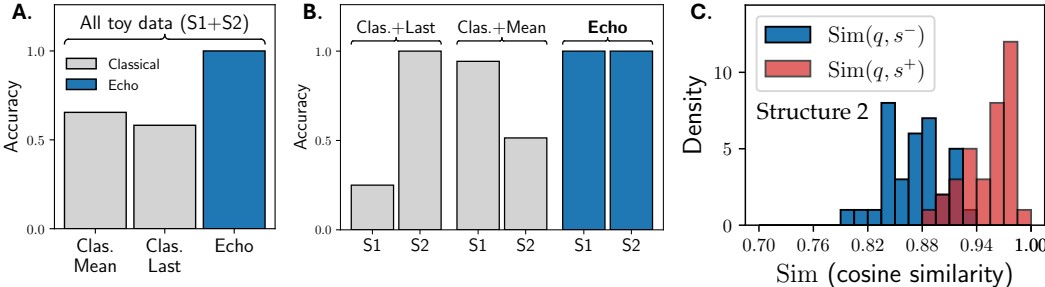

Figure 2: **A**. Accuracy of classical embeddings (mean and last-token pooling) and echo embeddings on the entire toy dataset. **B**. Accuracies of classical and echo embeddings split by the different structures of the data (S1 and S2, see text). **C**. We take the echo embeddings of only the first parts $A$ of query $q = [A, B]$ and sentences $s^-, s^+$ and plot the distribution of cosine similarities, showing that echo embeddings encode can later information in earlier tokens.

**Classical embeddings.** Traditionally, embeddings are computed by simply passing the sentence to the model and extracting some pooling (e.g. mean or last-token) of the contextualized embeddings corresponding to the input sentence. We will refer to embeddings created in this way as "*classical embeddings*". Additionally, one might first prompt the language model with an explanation of the task of interest followed by the sentence (Su et al., 2022).

## 3 ECHO EMBEDDINGS FROM LLMS

In this section, we describe our general methodology to extract high quality embeddings from autoregressive language models without bidirectional attention. We start with a simple synthetic dataset that demonstrates why causal attention might inhibit embeddings from capturing information reliably across the entire context (Section 3.1). We then present our proposed method of Echo Embeddings in Section 3.2, and test that our simple and intuitive approach in fact addresses the failure mode identified in the synthetic setup *without* introducing bidirectional attention.

### 3.1 FAILURE OF CLASSICAL EMBEDDINGS WITH CAUSAL ATTENTION

High-quality embeddings should effectively capture various text structures, including where similarity between inputs is strongly affected by either the early parts or the later parts of the inputs. We create a dataset with two "opposite" structures (S1 and S2) and show the classical embeddings with causal attention cannot capture both simultaneously.

The dataset we construct consists of a query $q$ and a similar sentence $s^+$ and a dissimilar sentence $s^-$. We measure the performance of embedding $\phi(\cdot)$ by computing the fraction where $\text{Sim}(q, s^+) > \text{Sim}(q, s^-)$. Each query and sentence is a concatenation of two strings, allowing us to vary whether early (or later) parts are similar (or discriminatory). Notationally, $A, A^+, B, B^+$ are semantically similar, while $A, A^-, B, B^-$ semantically dissimilar. We query GPT-4 to generate these various realistic but synthetic strings (see Section C).

**S1 (Early discriminatory; late redundant).** For a query $q = [A, B]$, in this structure, the early tokens provide information to discriminate between $s^+, s^-$ while the later tokens are all similar. In other words, $s^+ : [\mathbf{A^+}, B^+]$ and $s^- : [\mathbf{A^-}, B]$, where the discriminatory parts are bolded for convenience.

**S2 (Early redundant; late discriminatory).** Analogously, we consider a second "opposite" structure where the early tokens are all similar but the later tokens can discriminate between $s^+, s^-$. For query $q = [A, B]$, we have $s^+ : [A^+, \mathbf{B^+}]$ and $s^- : [A, \mathbf{B^-}]$, where the discriminatory parts are bolded for convenience.

We show the results of zero-shot classical embeddings extracted from Mistral-7B-Instruct-v0.1 in Figure 2 (A) on a dataset that has a mixture of both structures. We see that **neither mean nor last token pooling allows classical embeddings to achieve better-than-random performance**.

To get some insight into the failure, we plot the performance on each substructure separately in Figure 2 (B). **Last-token pooling disproportionately weights the later parts** of the sentence thereby missing discriminatory information in the early tokens in S1. In general, last-token pooling does not work well in a zero-shot manner on real data as well, as we see in the next section.

Mean-token pooling is better suited to the zero-shot setting and can allow us to extract information from all parts of the input. This works well in structure S1. However, the causal attention produces a different issue: the contextual embedding at position $k$, $\phi_k(x)$ *cannot* encode information about the later tokens $x_{k+1}, x_{k+2}, \ldots$. This causes **mean-token embeddings to exaggerate early token similarity and dilute discriminatory information in the later parts because of causal attention**. This effect is especially exacerbated when the early tokens are similar, as in S2. In total, classical embeddings fail when the dataset contains inputs of both instances—where the discriminatory information could be in the early or later tokens. We show that this failure mode of classical embeddings can be overcome in a completely zero-shot manner using a simple method which we describe next.

## 3.2 Echo Embeddings: A General Framework

We saw in Section 3.1 that while mean-token pooling aggregates information across all tokens, the contextualized representations of early tokens are suboptimal because they cannot attend to the later tokens due to the causal attention. Replacing the causal with bidirectional attention is a natural solution, and common intuition seems to be that this is necessary. We propose a new general embedding framework—echo embeddings—that captures contextualized information from all tokens in the input with only causal attention:

> **Echo embeddings method:** Prompt the language model to act as an autoencoder, e.g., by asking the model to {repeat, rephrase, fix, etc.} the input; feed the sentence $x$ to the language model *twice*; pool the contextualized embeddings of the *second* occurrence of $x$.
>
> Ex: "Rewrite the sentence: $x$; rewritten sentence: **x**"; embeddings are pooled from the bolded **x**.

The second occurrence can attend to everything, and hence extracting contextualized representations from the second occurrence *could* capture information from the entire input. Furthermore, in order to encourage the second occurrence to actually "encode" information about the first, we instruct the language model to perform a generic task that requires using this information, e.g., "rewrite" or "repeat". The exact prompt, template, or task used to extract embeddings can be varied to suit the model and downstream task. We find empirically that the performance of echo embeddings are not particularly sensitive to the exact phrasing of the prompt (Section 4.2). We provide a list of example prompts in Appendix C.

While echo embeddings is simple and intuitive, does it actually effectively capture bidirectional information?

We apply echo embeddings with mean-token pooling on the synthetic dataset introduced in Section 3, and present the results in Figure 2. Echo embeddings with mean-token pooling works significantly better than classical embeddings and **works well on *both* structures in the data—whether the discriminatory information is in the early or later tokens**.

In particular, let us focus on structure S2 where the discriminatory information appears in the later tokens. Echo embeddings with mean-token pooling work well, while classical embeddings did not because the early tokens could not attend to the later ones. This implies that echo embeddings can successfully address this failure and **the representations of early tokens can capture information "bidirectionally" about the entire input without changing causal to bidirectional attention**. To test this further, we measure whether echo embeddings can meaningfully distinguish $s^+$ and $s^-$ using just the information in the $A$ portions despite the discriminatory information solely appearing in the later $B-$ parts. Recall that echo embeddings extract embeddings from the second occurrence. We plot the cosine similarities $\mathrm{Sim}(q, s^+)$ and $\mathrm{Sim}(q, s^-)$ in Figure 2 (C). We find that $\mathrm{Sim}(q, s^+)$ is typically larger than $\mathrm{Sim}(q, s^-)$. Since we are only pooling the echo embeddings of the $A$ portion, any distinction between $s^+, s^-$ must come from the echo embeddings of $A$ capturing information from the later parts of the sentence.

Table 1: *Main results in the zero-shot setting:* Apples-to-apples comparison of zero-shot echo and classical embeddings with mean pooling, and other baselines on MTEB (56 datasets). For the autoregressive embeddings in this table, we extract embeddings from the Mistral-7B-Instruct-v0.1 model. Bolding only zero-shot models.

| Categories $\longrightarrow$ | Clas. | Clus. | P. Cls. | Rera. | Retr. | STS | Su. | **Average** |
|---|---|---|---|---|---|---|---|---|
| # of datasets | 12 | 11 | 3 | 4 | 15 | 10 | 1 | 56 |
| Echo (Ours) | 72.01 | 33.80 | **72.41** | 47.56 | 20.82 | **73.74** | 30.70 | 48.64 |
| Echo (compute matched) | 71.63 | 33.51 | 72.31 | 47.43 | **22.85** | 73.64 | **31.02** | **49.02** |
| Classical | 64.87 | **34.14** | 57.71 | 43.60 | 18.26 | 57.07 | 27.02 | 42.38 |
| PromptEOL | **73.84** | 27.53 | 55.47 | **48.44** | 13.18 | 67.14 | 28.33 | 43.69 |
| BERT | 61.47 | 29.48 | 56.33 | 43.44 | 10.44 | 52.98 | 29.82 | 37.87 |
| RoBERTa | 62.63 | 29.05 | 56.95 | 41.92 | 8.62 | 55.24 | 28.64 | 37.86 |
| LLM2Vec (*requires fine tuning*) | 72.51 | 40.06 | 70.95 | 50.43 | 19.74 | 71.90 | 27.84 | 49.43 |

**Summary.** We construct a synthetic dataset that showcases a failure mode in extracting embeddings from causal language models in the classical manner. While ours is a simplistic construction, we demonstrate that the failure mode we outline does occur on real data as well in Appendix C and could explain the poor performance of classical embeddings in practice. However, echo embeddings, where we repeat the input and take embeddings from the second occurrence, can successfully address this failure mode. We now test echo embeddings on a variety of real-world settings.

## 4    EXPERIMENTS

In this section, we describe how we implement and evaluate echo embeddings on real data.

### 4.1    MASSIVE TEXT EMBEDDING BENCHMARK

**MTEB & MTEB-MINI.** Our main evaluation dataset is the English-language subset of the Massive Text Embedding Benchmark (MTEB) (Muennighoff et al., 2022). MTEB is a collection of 56 datasets that are grouped into different embedding tasks: classification, clustering, pair classification, reranking, retrieval, sentence similarity (STS), and summarization, with the goal of evaluating embeddings broadly. However, as the name suggests, MTEB is *massive*, and requires multiple days to evaluate a 7B-parameter model on 8 A100 GPUs. Thus, we adopt a smaller 28-dataset subset of MTEB that spans all categories except summarization, which we call *MTEB-MINI*. We provide a detailed description of MTEB and MTEB-MINI in Appendix A.1.

### 4.2    EVALUATION OF ZERO-SHOT EMBEDDINGS

We begin with the zero-shot setting, where embeddings are extracted from a pre-trained model without additional training. Our goal is to show that without any additional training, echo embeddings outperform prior entirely-zero-shot methods, and can perform similarly to unsupervised approaches (which require additional fine-tuning with unsupervised data), such as LLM2Vec-unsupervised (BehnamGhader et al., 2024). While supervised fine-tuning is often used to improve embedders, zero-shot embeddings have the advantage that no further (potentially expensive) training is required.

**Constructing zero-shot echo embeddings.** Following the echo embeddings method described in Section 3, to construct echo embeddings we need to pick a prompt, a pooling strategy, and a language model. We evaluate the impact of these choices on the performance of the embeddings on the Massive Text Embedding Benchmark (MTEB) (Muennighoff et al., 2022).

*Choosing a prompt:* As specified in Section 3, we must prompt the language model to act as an autoencoder. Sclar et al. (2023) argues that on many tasks, the exact formatting and wording of the prompt can strongly influence performance. Thus, we investigate: *does the choice of prompt affect embeddings quality?* As we shall discuss later, we find that, **the choice of prompt does not substantially affect the performance of echo embeddings** (Section 4.3). This implies that prompt

Table 2: *Role of scale and base model in the zero-shot setting*: Average MTEB score (56 datasets) for LLaMA-7B-2-Instruct and S-LLaMA-1.3B.

| Model | Echo | Classical | PromptEOL |
|---|---|---|---|
| S-LLaMA-1.3B | **40.45** | 34.31 | 37.50 |
| LLaMA-7B | **45.84** | 40.29 | 42.84 |

Table 3: *Casting to bidirectional attention:* We evaluate MTEB-MINI (28 datasets) and compare classical with causal attention (Cl.+Uni) and with bidirectional attention (Cl.+Bi).

| Model | Echo | Cl.+Uni | Cl.+Bi |
|---|---|---|---|
| LLaMA-2-7B | **56.99** | 47.27 | 43.03 |
| S-LLaMA-1.3B | **49.55** | 40.14 | 35.77 |
| Mistral-7B | **59.78** | 49.03 | 58.24 |

tuning is unnecessary, and *any* reasonable prompt will perform well. For our main results, we adopt the prompt "Rewrite the following paragraph: $S$. The rewritten paragraph: $S$" for echo embeddings and "Write a paragraph: $S$" for classical embeddings.

*Choosing a pooling strategy:* We evaluate both mean and last-token pooling for both classical and echo embeddings 4. Consistent with our toy-data analysis in Section 3, we find that mean token pooling is required for good embeddings and thus use this for all further zero-shot experiments (Table 4).

*Choosing a language model:* We evaluate the performance of echo embeddings on three language models: Mistral-7B-Instruct-v0.1 (Jiang et al., 2023a), LLaMA-2-7B-Instruct (Touvron et al., 2023a), and S-LLaMA-1.3B (Xia et al., 2023) (sometimes abbreviated). We select the instruction-finetuned model for each of them. Refer to Appendix A.2 for additional information on the base models.

Table 4: *Role of pooling in the zero-shot setting:* Average MTEB Score (56 datasets) for mean and last token pooling.

| | Mean | Last-token |
|---|---|---|
| Echo | **48.64** | 31.55 |
| Classical | **42.38** | 31.94 |

*Compute-matching echo embeddings.* Naively, echo embeddings would require twice the compute of classical embeddings. To account for this, we also consider compute-matched echo embeddings in which we halve the length of the input.

**Alternative approaches to constructing embeddings from autoregressive models.** In addition to our results, we compare to two recently proposed methods to convert autoregressive models into embedders without supervised fine-tuning: PromptEOL (zero-shot) and the unsupervised variant of LLM2Vec (requires additional fine-tuning). PromptEOL prompts the model to summarize the input in a single token, and extracts embeddings from the predicted token (Jiang et al., 2023b) (see Appendix D for additional details). The unsupervised variant of LLM2Vec (LLM2Vec-unsupervised) incorporates an additional masked-language-modeling fine-tuning step. Importantly, LLM2Vec is *not* zero-shot because it involves this extra fine-tuning step.

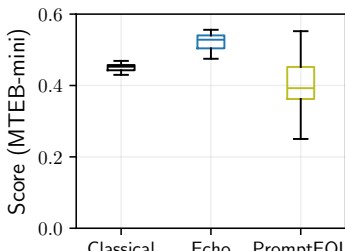

Figure 3: *Role of prompt in the zero-shot setting:* Variance of zero-shot performance across different prompts and formats on MTEB-MINI (28 datasets). Refer to Appendex D for details on the exact prompts and formats.

### 4.3 ZERO-SHOT RESULTS

**Echo embeddings yield strong embeddings, and without additional compute.** We present the main zero-shot results in which we evaluate echo embeddings on MTEB in Table 1. We find that echo embeddings successfully converts causal language models into strong encoders. For Mistral-7B-Instruct, we find, surprisingly, that compute-matched echo embeddings, achieves slightly *higher* performance than full-compute echo embeddings. The difference is small enough that it is unclear to what degree we would observe this improvement in other settings, but suggests that echo embeddings can be adapted to outperform classical embeddings without any additional compute. We extend this analysis of compute in Section 4.6. We also evaluate echo embeddings over multiple models and scales in Table 2 and find consistent results.

Table 5: *Main fine-tuning results:* We report MTEB scores for echo embeddings with full compute and compute-matched and classical embeddings with a causal and with a casted bidirectional architecture. For echo embeddings, we use Mistral-7B as the backbone model, and last-token pooling. For the compute matched echo embeddings, FT+IT refers to both fine-tuning and inference-time compute matching.

| Categories ⟶ | Clas. | Clus. | P. Cls. | Rera. | Retr. | STS | Su. | **Average** |
|---|---|---|---|---|---|---|---|---|
| # of datasets | 12 | 11 | 3 | 4 | 15 | 10 | 1 | 56 |
| Echo (ours) | **77.43** | **46.32** | 87.34 | **58.14** | **55.52** | 82.56 | 30.73 | **64.68** |
| Echo (FT+IT compute matched) | 77.39 | 46.27 | 87.49 | 58.08 | 55.07 | **83.18** | 30.73 | 64.66 |
| Classical | 76.57 | 45.78 | 86.37 | 56.71 | 54.87 | 82.03 | **31.02** | 63.98 |
| Classical + bidirectional attn. | 76.70 | 45.94 | **88.15** | 57.23 | 54.96 | 82.42 | 29.32 | 64.23 |

**Comparison to PromptEOL.** We find that echo embeddings outperform PromptEOL on average across the tested MTEB datasets by 5%. In addition to this improvement, echo embeddings is additionally much less sensitive to the exact wording and formatting of the prompt (Figure 3).

**Comparison to prior encoder models.** We evaluate the zero-shot MTEB performance of existing bidirectional (encoder) models, BERT-large and RoBERTa-large, which are often used as the backbone of fine-tuned embedders. We find that echo embeddings outperform both BERT-large and RoBERTa-large in the zero-shot setting. We suspect that the performance improvement is due to Mistral-7B being significantly more powerful than BERT and RoBERTa. This highlights the importance of our method: by using echo embeddings, we can leverage the full power of autoregressive language models, which are generally dominant over bidirectional models on other NLP tasks.

**Comparison to LLM2Vec.** We find that *entirely zero-shot* echo embeddings can perform similarly to LLM2Vec-unsupervised, despite the fact that LLM2Vec-unsupervised is *not* a zero-shot method, and requires additional fine-tuning on top of the pre-trained base model. Echo embeddings thus offers a simple alternative to LLM2Vec that does not require additional training.

**Relaxing to bidirectional attention.** We find that changing the attention mask from causal to bidirectional often substantially underperforms the causal architecture, but not for Mistral-7B (Table 3). Regardless, echo embeddings consistently outperforms this baseline. A priori, we might expect that, without further fine-tuning, such a modification to the architecture would harm performance, as we would expect that the model has not been trained to accommodate the additional attention. For LLaMA-2-7B and S-LLaMA-1.3B, this is the case. However, for Mistral-7B, casting to bidirectional attention does not harm performance and instead yields performance similar to echo embeddings. We suspect that this may arise from non-standard pre-training methodology, as discussed by BehnamGhader et al. (2024).

**Ablations.** We perform multiple ablations to evaluate the role of each component:

1. We test the role of pooling strategy in Table 4. Consistent with our analysis in Section 3, we find that mean pooling outperforms last-token pooling by a wide margin for both echo and classical embeddings. This suggests that mean pooling is required for echo embeddings in the zero-shot setting.
2. We evaluate the role of the exact wording and formatting of the prompt in Figure 3. We find that the performance of echo embeddings has low variance with respect to the exact choice of prompt. This implies that echo embeddings does not require prompt tuning to perform well. Further, we find that *all* tested echo embeddings prompts outperform every tested classical embeddings prompt.

## 4.4 EVALUATION OF FINE-TUNED EMBEDDINGS

Echo embeddings also perform well in the fine-tuning setting. We will show that echo embeddings can outperform classical embeddings, even when the attention is converted to be bidirectional. Further, echo embeddings can match the performance of other methods that require much more complicated fine-tuning methodology, with identical fine-tuning and inference-time compute, and without modifying the architecture.

Table 6: *Role of base model in the fine-tuning setting*: We compare echo and classical embeddings for S-LLaMA-1.3B. FT+IT refers to the setting in which echo embeddings are compute-matched at train and inference time.

| Methods | MTEB Score |
|---|---|
| Echo | **62.01** |
| Echo (FT+IT compute matched) | 61.65 |
| Classical | 61.26 |

Table 7: *Comparisons to recent baselines*: MTEB score for models trained on similar data. Echo embeddings matches the performance of models trained on similar data, but without requiring bidirectional attention. FT+IT refers to the setting in which echo embeddings are compute-matched at train and inference time.

| Method | MTEB Score |
|---|---|
| Echo (FT+IT compute matched) | 64.66 |
| LLM2Vec-supervised | 64.80 |
| GritLM-7B (public data) | 64.70 |

**Constructing fine-tuned echo embeddings.** To construct fine-tuned embeddings, we must again choose a prompt, a pooling strategy, and a language model. In addition, we need to choose a contrastive learning objective and a collection of training datasets.

*Choosing a prompt:* We adopt a minimal prompt that differs for queries and documents. Prior work has shown that the addition of an instruction can improve the quality of the embeddings (Wang et al., 2023), and thus we include an instruction in the prompt. The prompts are:

| *Echo (queries)* | *Echo (docs)* | *Classical (queries)* | *Classical (docs)* |
|---|---|---|---|
| Instruct: {instruction} | Document: $S$ | Instruct: {instruction} | Document: $S$ |
| Query: $S$ | Document again: $S'$ | Query: $S$ | |
| Query again: $S'$ | | | |

We provide a list of the instructions in Appendix F.

*Choosing a pooling strategy:* We evaluate both mean and last-token pooling for both classical and echo embeddings. In practice, we find that last-token pooling performs slightly better than mean pooling in this setting (see Section 4.5).

*Choosing a language model:* We train and evaluate the performance of echo embeddings on Mistral-7B-Instruct-v0.1 and S-LLaMA-1.3B (Jiang et al., 2023a; Xia et al., 2023). We choose Mistral-7B as the best-performing model in the zero-shot setting, and S-LLaMA-1.3B to test the broad applicability of our method across model class and scale.

*Training datasets and optimization:* We train on a collection of publicly available datasets that are standard training datasets in the embedding literature. We list and describe each of the datasets in Appendix F. To fine-tune the model, we optimize the SimCSE loss with in-batch and mined hard negatives. Since this is standard, we defer discussion of this to Appendix F. Each batch is constructed by sampling a dataset from our set of training dataset, and then collecting examples from only this dataset. We use GradCache to train with a large batch size (2048) with limited GPU memory (Gao et al., 2021a). We train with LoRA instead of full finetuning, with $r = 16$ and $\alpha = 16$. We choose $\tau = 1/50$ and a learning rate of $8 \times 10^{-4}$.

*Compute-matching echo embeddings.* Echo embeddings require twice the compute of classical embeddings. Similar to the zero-shot setting, we reduce the inference-time cost of echo embeddings by **halving the length of the input.** However, in the training setting, we additionally must match the train compute. We opt to **train for half as many steps** as classical embeddings, as this is the most straightforward way to match the compute. The compute-matched echo embeddings in Table 5 refer to the setting where we match compute for both training and inference.

**Classical embeddings with bidirectional attention.** In addition to our comparison to classical embeddings with causal attention, we train a model for which we replace the causal attention with bidirectional attention.

## 4.5 FINE-TUNING RESULTS

**Echo embeddings yield high quality embeddings in the fine-tuning setting.** We present the main fine-tuning results in Table 5. Similar to the zero-shot setting, compute-matched echo embeddings

outperform classical embeddings with both architectures: causal and bidirectional. When additional compute is available, echo embeddings improve performance further. This suggests that the fundamental gap between classical and echo embeddings that we identified in Section 3 persists even after fine-tuning. We find consistent results when evaluating across model classes and scales (Table 6).

**Comparison to LLM2Vec.** We compare to the supervised variant of LLM2Vec. The LLM2Vec method relaxes the attention of the base autoregressive model to bidirectional and requires two fine-tuning stages: (1) fine-tuning with an unsupervised masked-language-modeling objective (2) supervised fine-tuning for embeddings. Despite the complexity of the LLM2Vec method and additional fine-tuning, echo embeddings nonetheless approximately matches the performance with identical fine-tuning and inference-time compute (Table 7). Echo embeddings thus offers itself as a much simpler method that does not require modifying the architecture nor additional fine-tuning stages, with nearly identical performance.

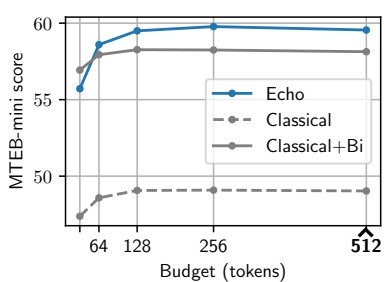

Figure 4: *Zero-shot setting*: MTEB-mini score of zero-shot echo embeddings and classical embeddings with bidirectional attention with fixed compute (token) budget. We use the zero-shot Mistral 7B from Table 1. The standard token budget for MTEB is 512.

**Comparison to GritLM.** We also compare to GritLM (Muennighoff et al., 2024) in Table 7. This approach, which simultaneously trains an embedder and a generative model, finds the bidirectional attention relaxation to be crucial. We find that compute-matched echo embeddings perform similarly the variant of GritLM that has been trained on similar data as ours (public data), even though echo embeddings use a causal architecture.

**The role of pooling.** We investigate the role of pooling in Appendix F. In the fine-tuning setting, the choice of pooling strategy has a much weaker effect than in the zero-shot setting. Consistent with prior work (Wang et al., 2023; Zhang et al., 2023a), we find that last-token pooling, with the addition of a trainable end-of-sentence token embedding, slightly outperforms mean pooling.

## 4.6 ADDRESSING THE INFERENCE-TIME COST OF ECHO EMBEDDINGS

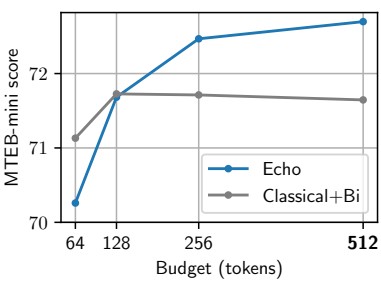

One potential drawback of echo embeddings is the computational cost of the method at both training and inference time. Naively, echo embeddings requires approximately twice the compute of classical embeddings, as the input is repeated. We have so far seen that in an entirely compute-matched training and inference time setting, echo embeddings outperforms classical embeddings (Table 5). In this section, we expand this analysis to consider how echo embeddings perform under different inference-time compute budgets. We explore the role of training under different budgets in Appendix F.

Figure 5: *Fine-tuning setting*: MTEB-mini score of fine-tuned echo embeddings and classical embeddings with bidirectional attention with fixed compute (token) budget. We use the trained Mistral-7B models from Table 5. The standard token budget for MTEB is 512.

## 4.7 THE ROLE OF COMPUTE

**The role of inference-time compute in the zero-shot setting.** In Section 4.2, we demonstrated that by halving the number of input tokens that are encoded by echo embeddings, thus matching compute with the classical approach to emebddings, we achieve stronger performance than classical embeddings. Here, we investigate the role of compute in further detail by plotting the performance of echo embeddings as a function of the total number of tokens encoded

(Figure 4). We find that when the budget is greater than approximately 64 tokens, echo embeddings outperforms Mistral-7B classical embeddings with bidirectional attention. For very small computational budgets (fewer than 64 tokens), we find that Mistral-7B classical embeddings with bidirec-

tional outperform echo embeddings. Since the standard budget for MTEB is 512 tokens, echo embeddings likely offer stronger performance in most settings. Over all compute budgets, echo embeddings outperform causal attention classical embeddings.

**The role of inference-time compute in the fine-tuning setting.** The fine-tuning setting exhibits a similar trend: for very small computational budgets, echo embeddings perform worse than classical embeddings, but as the budget increases echo embeddings outperform classical embeddings (Figure 5). In the fine-tuning setting, echo embeddings outperform classical embeddings for budgets greater than 128 tokens, which is still much smaller than the standard MTEB budget (512 tokens).

## 5 RELATED WORK

**Sentence embeddings.** Dense low-dimensional vectors representing textual semantics has been widely studied and applied. Early approaches involved computing embeddings for individual words (Hinton, 1984; Rumelhart et al., 1986; Elman, 1990; Mikolov et al., 2013; Pennington et al., 2014). Later work aims to compute dense vectors representing the semantics of entire sequences by combining or composing word vectors (Le & Mikolov, 2014; Iyyer et al., 2015; Kiros et al., 2015; Socher et al., 2011; Tai et al., 2015; Wang et al., 2016; Wieting et al., 2015). Khattab & Zaharia (2020) propose to use late interaction between document and query vectors to improve retrieval performance. Reimers & Gurevych (2019) propose S-BERT which takes a pretrained BERT (Devlin et al., 2018) and trains with a triplet loss. More recent approaches typically adopt this approach to different pretrained models (Raffel et al., 2020; Ni et al., 2021b; Raffel et al., 2020) and contrastive objectives (Oord et al., 2018; Gao et al., 2021b). Other papers propose including prompts to improve task-specific embedding performance (Jiang et al., 2022; Su et al., 2022). Some work combines multiple of these training objectives and approaches (Xiao et al., 2023a; Li et al., 2023). Most recent sentence embeddings research has focused on improving finetuning. Reimers & Gurevych (2019) demonstrates that without finetuning, BERT has low-quality embeddings. To our knowledge, Jiang et al. (2023b) is the only paper that constructs zero-shot embeddings for autoregressive language models.

**Next-token language modeling for embeddings.** A series of papers aim to construct high quality embeddings from autoregressive large language models. Most closely related, BehnamGhader et al. (2024) proposed LLM2Vec which adds an additional unsupervised training step on a masked-language-modeling objective.Multiple papers apply the fine-tuning approach of S-BERT to language models but using a trained GPT (Radford et al., 2018) as the backbone architecture (Muennighoff, 2022; Zhang et al., 2023a). Ma et al. (2023) adopts this approach but for LLaMA-2 (Touvron et al., 2023b). Jiang et al. (2023b) extracts embeddings by asking a language model to summarize the input sentence. Wang et al. (2023) improves embeddings by adding synthetic training data and trains on Mistral (Jiang et al., 2023a). Muennighoff et al. (2024) adds an additional objective to maintain generation performace, but requires bidirectional attention. Li & Li (2023b) modifies the architecture of a causal language model to be bidirectional.

**Language modeling with bidirectional information.** Xu et al. (2023) finds that repetition can improve the performance of autoregressive language models for reasoning tasks. Arora et al. (2024) finds that repetition can be useful to improve recall for recurrent architectures.

## 6 CONCLUSION

We have demonstrated that the common assumption that bidirectional architectures are crucial for high quality embedding models is false. Our method, echo embeddings, prompts an autoregressive model to act as an autoecoder, which enables the model to encode bidirectional information. Our results highlight the importance of bidirectional information—we find that the inability to encode bidirectional information leads to the failure of classical-style embeddings in autoregressive models. Further, our method demonstrates that by carefully choosing the task that the language model is prompted to perform *at inference time* is a promising approach for improving embeddings. We hope that our work will inspire future research towards understanding how inference-time changes can impact embeddings.

## ACKNOWLEDGMENTS

This material is based upon work supported by the National Science Foundation Graduate Research Fellowship under Grant No. DGE2140739. Any opinion, findings, and conclusions or recommendations expressed in this material are those of the authors(s) and do not necessarily reflect the views of the National Science Foundation.

This research was supported by the Center for AI Safety Compute Cluster. Any opinions, findings, and conclusions or recommendations expressed in this material are those of the author(s) and do not necessarily reflect the views of the sponsors.

This work was supported in part by the AI2050 program at Schmidt Sciences (Grant #G2264481).

We gratefully acknowledge the support of Apple.

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

## A  GENERAL INFORMATION FOR REPRODUCIBILITY

In this section we include information that might aid in reproducibility that is not specific to any specific setting in the paper.

### A.1  MASSIVE TEXT EMBEDDING BENCHMARK

The Massive Text Embedding Benchmark (MTEB) is a collection of datasets from seven categories: classification, clustering, pair classification, reranking, retrieval, sentence similarity (STS), and summarization. The leaderboard is published at `https://huggingface.co/spaces/mteb/leaderboard`. The list of datasets and their descriptions can be found at Muennighoff et al. (2022) in Appendix A.

### A.2  BASE MODEL HUGGINGFACE IDS

In this paper, we use the following models:

- Mistral 7B instruction-tuned: mistralai/Mistral-7B-Instruct-v0.1
- LLaMA 7B instruction-tuned: meta-llama/Llama-2-7b-chat-hf
- S-LLaMA 1.3B: princeton-nlp/Sheared-LLaMA-1.3B
- LLaMA 13B instruction-tuned: meta-llama/Llama-2-13b-chat-hf

## B    SOCIETAL CONSIDERATIONS

In this section we discuss the societal impact of our work and the safeguards we have taken to mitigate negative impacts.

### B.1    IMPACT

We intend for and believe that this work will have a positive societal impact. In particular, language models are known to suffer from limitations such as hallucinations and biases. In the context of retrieval-augmented generation, we believe that higher quality embeddings can lead to more accurate retrievals, which can in turn lead to more accurate generations. This can help mitigate the impact of hallucinations and biases in language models. We also believe that the improved embeddings can be used in a variety of downstream tasks, such as information retrieval, question answering, and summarization, which can have a positive impact on society. Nonetheless, we acknowledge that not all uses of our work will be necessarily positive. For example, our work can be used to improve the performance of language models in generating fake news or misinformation. Second, we release a training dataset that has been aggregated from a set of other open source datasets that might nonetheless contain harmful content. We believe that the positive impacts of our work outweigh the negative impacts, and we are committed to working towards ensuring that our work is used for positive purposes.

## C    ECHO EMBEDDINGS: ADDITIONAL INFORMATION

In this section, we aim to describe the additional details that were omitted from Section 2 and 3.

**Cosine Similarity.**    As discussed in Section 2, we often use the cosine similarity to measure the similarity in embeddings. Recall that given two sentences $x$ and $y$, we wish to determine the degree to which they are semantically similar. Cosine similarity,

$$\text{Sim}(x, y) := \frac{\langle \phi(x), \phi(y) \rangle}{\|\phi(x)\| \|\phi(y)\|}, \tag{1}$$

measures the similarity between the embeddings of $x$ and $y$ for any embedding function $\phi \colon \mathcal{X} \to R^d$. The cosine similarity is used for our experiments in Sections 3, and as the similarity function for training in 4.4. All MTEB datasets use cosine similarity to compute similarity with the exception of the classification datasets, in which similarity is not explicitly measured, and the clustering datasets, which use Euclidean distance,

$$\text{Sim}(x, y) := \|\phi(x) - \phi(y)\|, \tag{2}$$

as a metric.

**Prompts for Section 3.**    For these experiments, we only evaluate with a single prompting strategy. For classical embeddings, we encode a sentence $S$ using the prompt:

$$x = \text{Write a sentence: } S$$

We take the pooled embedding to be the mean token embedding $\phi_S(x)$. For echo embeddings, we encode a sentence $S$ using the prompt:

$$x = \begin{array}{l} \text{Rewrite the following sentence: } S \\ \text{The rewritten sentence: } S' \end{array}$$

where $S' = S$ and we let our pooled embedding be the mean token embedding $\phi_{S'}(x)$. We do not evaluate with the last-token pooling strategy in this Section.

**General Prompting Guidelines.**    Throughout the paper, we use a variety of different prompts to construct embeddings. In Section D, we demonstrate that for zero-shot embeddings, the exact wording or template used as a prompting strategy does not have a strong effect on the performance of MTEB tasks, with the exception of for the summarization approach. This implies, in general, that classical embeddings and echo embeddings should be robust to the exact choice of prompts. The

important component of echo embeddings is instead the structure: the input text should be repeated twice when computing embeddings, and the embeddings should be taken over the second occurrence of the input text.

**Example classical embedding structures:**

Say the sentence: $S$

Write the phrase: $S$

Complete the query: $S$

Explain the text: $S$

**Example echo embedding structures:**

Repeat the sentence: $S$
The sentence again: $S'$

Rephrase the query: $S$
The query rephrased: $S'$

Fill in the blank: $S$
The blanks filled in: $S'$

Rewrite the text: $S$
The sentence rewritten: $S'$

**Toy data.** We provide a subset of the toy data from Section 2. For Structure 1, the data is given in Table 8. For Structure 2, the data is given in Table 9. For Structure 3, the data is given in Table 10. In all cases, the data is generated by GPT-4. The data from Structure 1 is generated from the following GPT-4 prompt, and the other structures are generated from minor variations on this:

Together, we need to generate sentence triplets. Each triplet will have the following form:
- Sentence 1 can be anything, be creative here.
- Sentence 2 must represent something opposite to sentence 1, however, it is important that the first half of the sentence is exactly the same as the first half of sentence 2. The only difference in wording can be in the second half of the sentence.
- Sentence 3 should be extremely similar to sentence 1 and semantically equivalent, but slightly reworded.

Here is an example:
{
"sentence1": "I like to eat apples and bananas but I really hate almost every other fruit.",
"sentence2": "I like to eat apples and bananas and I also enjoy also every other fruit",
"sentence3": "I like two fruits: apples and bananas but I hate nearly all fruits other than these.",
}

The first half of the sentence should be relatively short, less than 10 words, but the second half should be long, at least 10 words. Give more examples, and write them in json format. Be creative!

| $q$ | $s^-$ | $s^+$ |
|---|---|---|
| She loves to travel in summer, especially to cold destinations, avoiding hot and crowded places | She loves to travel in summer, but prefers to visit hot and bustling tourist spots | In summer, she adores traveling, specifically to chilly locations, steering clear of warm, populous areas |
| The cat often sits by the window, dreaming of chasing birds and enjoying the warm sunshine | The cat often sits by the window, but is too lazy to dream of chasing anything | Frequently, the cat lounges near the window, imagining bird pursuits and basking in the sunlight |
| He reads books every night, finding solace in fiction and escaping from the stresses of daily life | He reads books every night, yet he feels that non-fiction is more engaging and informative | Nightly, he immerses himself in books, seeking comfort in stories and evading everyday tensions |
| They play music loudly in the evening, filling their home with energetic beats and vibrant melodies | They play music loudly in the evening, but only soothing classical tunes to relax | In the evenings, they blast tunes, their house resonating with lively rhythms and bright harmonies |
| She paints landscapes on weekends, expressing her creativity through vibrant colors and abstract forms | She paints landscapes on weekends, preferring realistic and detailed depictions of nature | On weekends, she engages in landscape painting, showcasing her artistic flair with lively hues and unconventional shapes |
| The children eagerly await winter, dreaming of snowball fights and building snowmen | The children eagerly await winter, yet they dislike the cold and prefer staying indoors | During winter, the kids are excited, imagining snow battles and constructing snow figures |
| He often jokes at parties, becoming the center of attention with his witty humor | He often jokes at parties, but tends to alienate others with his sarcasm | At social gatherings, he frequently makes jokes, captivating the crowd with his clever wit |
| She collects antique vases, adoring their unique designs and historical significance | She collects antique vases, but is indifferent to their history and focuses on their resale value | Her hobby is gathering old vases, cherishing their distinct patterns and the stories they hold |
| The band plays rock music loudly, thrilling audiences with energetic performances and powerful lyrics | The band plays rock music loudly, but often receives complaints for being too noisy | Performing rock loudly, the band excites crowds with dynamic shows and impactful words |
| He prefers working at night, enjoying the quiet and focusing better without distractions | He prefers working at night, despite feeling more tired and less productive | Nighttime is his preferred work period, appreciating the tranquility and concentrated environment |
| She writes poetry in her free time, pouring her emotions and experiences into each verse | She writes poetry in her free time, but struggles to find inspiration and motivation | During her leisure, she crafts poems, infusing her feelings and life stories into every line |

Table 8: Examples of Structure 1 from Section 3

| $q$ | $s^-$ | $s^+$ |
|---|---|---|
| On sunny days, I often find myself longing for the cool breeze of the ocean and the sound of waves crashing, as I enjoy outdoor activities | During rainy days, I usually prefer the warmth and quiet of my home, as I enjoy outdoor activities | When the sun is shining, I tend to crave the refreshing sea air and the rhythmic sound of the ocean, since I relish spending time outdoors |
| As a lover of classical music, I spend hours listening to Beethoven and Bach, reveling in the complexity of their compositions, though I'm fond of playing the guitar | Despite my preference for rock music, I rarely spend time on music other than playing my favorite tunes on the guitar, though I'm fond of playing the guitar | Being an enthusiast of classical melodies, I often indulge in lengthy sessions of Beethoven and Bach, appreciating the intricacies of their work, as I delight in guitar playing |
| In the world of literature, I have an insatiable appetite for mystery novels and spend countless evenings unraveling their plots, but I adore reading poetry | Contrary to my usual tastes, I rarely delve into mystery novels and prefer lighter reading materials, but I adore reading poetry | As a fervent reader, my passion lies in the twists and turns of mystery stories, which I often explore during long nights, yet I cherish reading poetry |
| Growing up in a bustling city, I've always been surrounded by the constant hum of activity and the bright city lights, which makes me appreciate quiet countryside walks | Having been raised in a tranquil rural area, I'm more accustomed to the sounds of nature and open fields, which makes me appreciate quiet countryside walks | Raised in the lively atmosphere of an urban environment, I'm used to the never-ending city noise and glowing nights, leading me to enjoy the serenity of rural strolls |
| Ever since I was a child, fascinated by the vastness of the universe, I would spend countless nights gazing at the stars through my telescope, dreaming of exploring distant galaxies, yet I still find solace in simple nature hikes | Though I've always been more interested in the immediate world around me, preferring to focus on the tangible and the present, I rarely look up at the night sky, yet I still find solace in simple nature hikes | From my early years, captivated by the infinity of space, I devoted many nights to stargazing and imagining interstellar journeys, but I also enjoy the peace of nature walks |
| Growing up with a passion for culinary arts, experimenting with exotic ingredients and complex recipes, and often spending whole days in the kitchen perfecting new dishes, I also have a deep appreciation for classic literature | Despite my lack of interest in cooking and a preference for simple, quick meals that require minimal preparation, I'm not one to spend time in the kitchen, I also have a deep appreciation for classic literature | Since childhood, I've been enthusiastic about cooking, often trying out unusual ingredients and intricate recipes, dedicating entire days to refining my culinary creations, and I equally cherish classic literary works |

Table 9: Examples of Structure 2 from Section 3

| $q$ | $s^-$ | $s^+$ |
|---|---|---|
| SShe loves to travel in summer, especially to cold destinations, avoiding hot and crowded places | She loves to travel in summer, but prefers to visit hot and bustling tourist spots | She loves to travel in summer, specifically to chilly locations, steering clear of warm, populous areas |
| The cat often sits by the window, dreaming of chasing birds and enjoying the warm sunshine | The cat often sits by the window, but is too lazy to dream of chasing anything | The cat often sits by the window, imagining bird pursuits and basking in the sunlight |
| He reads books every night, finding solace in fiction and escaping from the stresses of daily life | He reads books every night, yet he feels that non-fiction is more engaging and informative | He reads books every night, seeking comfort in stories and evading everyday tensions |
| They play music loudly in the evening, filling their home with energetic beats and vibrant melodies | They play music loudly in the evening, but only soothing classical tunes to relax | They play music loudly in the evening, their house resonating with lively rhythms and bright harmonies |
| She paints landscapes on weekends, expressing her creativity through vibrant colors and abstract forms | She paints landscapes on weekends, preferring realistic and detailed depictions of nature | She paints landscapes on weekends, showcasing her artistic flair with lively hues and unconventional shapes |
| The children eagerly await winter, dreaming of snowball fights and building snowmen | The children eagerly await winter, yet they dislike the cold and prefer staying indoors | The children eagerly await winter, imagining snow battles and constructing snow figures |
| He often jokes at parties, becoming the center of attention with his witty humor | He often jokes at parties, but tends to alienate others with his sarcasm | He often jokes at parties, captivating the crowd with his clever wit |
| She collects antique vases, adoring their unique designs and historical significance | She collects antique vases, but is indifferent to their history and focuses on their resale value | She collects antique vases, cherishing their distinct patterns and the stories they hold |
| The band plays rock music loudly, thrilling audiences with energetic performances and powerful lyrics | The band plays rock music loudly, but often receives complaints for being too noisy | The band plays rock music loudly, the band excites crowds with dynamic shows and impactful words |
| He prefers working at night, enjoying the quiet and focusing better without distractions | He prefers working at night, despite feeling more tired and less productive | He prefers working at night, appreciating the tranquility and concentrated environment |
| She writes poetry in her free time, pouring her emotions and experiences into each verse | She writes poetry in her free time, but struggles to find inspiration and motivation | She writes poetry in her free time, infusing her feelings and life stories into every line |

Table 10: Examples of Structure 3 from Section 3

| Most improved | | | Least improved | | |
|---|---|---|---|---|---|
| Sentence 1 | Sentence 2 | Score | Sentence 1 | Sentence 2 | Score |
| The best thing you can do is to know your stuff. | The best thing to do is to overcome the fussiness. | 0.0 | Sometime if you really want it you might need to pay an agency to get the place for you. | You could probably get a tour agency to do it for you but it would cost you. | 2.0 |
| It really doesn't matter. | It doesn't matter unless it is really far off. | 3.0 | There are three options: | There are only three options: | 5.0 |
| I think it's fine to ask this question. | I think it is okay to ask the question. | 5.0 | Bremer said one initiative is to launch a US$70 million nationwide program in the next two weeks to clean up neighborhoods and build community projects. | Bremer said he would launch a $70-million program in the next two weeks to clean up neighborhoods across Iraq and build community projects, but gave no details. | 3.6 |
| What kind of insulation is it? | What kind of floors are above? | 0.0 | "Tony's not feeling well," Spurs coach Gregg Popovich said. | We're thrilled to be up 3-2," Coach Gregg Popovich said Wednesday. | 1.6 |
| It depends entirely on your company and your contract. | I guess it depends on the nature of your contract. | 4.0 | Shares of Mandalay closed down eight cents to $29.42, before the earnings were announced. | Shares of Mandalay closed down 8 cents at $29.42 Thursday. | 4.0 |
| You need to read a lot to know what you like and what you don't. | You have to know what you want to do. | 0.0 | Singapore reported no suspected SARS cases Wednesday, but officials quarantined 70 people who had contact with the Taiwanese patient. | Still, Singapore quarantined 70 people who had been in close contact with the scientist. | 3.0 |
| I would say you can do it, but it wouldn't be advised. | Personally, I would say not unless it suits you. | 2.0 | The dollar was at 117.85 yen against the Japanese currency, up 0.1 percent. | Against the Swiss franc the dollar was at 1.3289 francs, up 0.5 percent on the day. | 1.333 |

Table 11: Example sentences from STSBenchmark in which zero-shot echo embeddings with Mistral 7B most improve **(left)** and least improve **(right)**.

## D    ADDITIONAL ZERO-SHOT RESULTS

Recent literature suggests that the performance of language models on zero-shot tasks can be highly variable depending on the exact wording and template of the prompts (Sclar et al., 2023). Thus, in order to test the degree to which the exact prompt matters for each of the embedding strategies we consider, we perform *prompt randomization* where we sample prompts by randomizing the exact wording, punctuation, and capitalization of the prompt.

**Prompt sampling procedure.**    Here we describe the prompt sampling procedure and then provide the prompts that we use for the zero shot:

1. Choose an instruction. For classical embeddings, we choose from {Write, Say, Complete, Explain}. For echo embeddings, we choose from {Repeat, Rewrite, Rephrase, Fill in the blank}. For PromptEOL, we choose from {Summarize, Categorize, Understand, Analyze}.

2. Choose a wording for the instruction. For example, if we chose "Say" as the instruction, then we would choose from {Say a sentence, Say a paragraph, Say something, Say a response, Say a query, Say a prompt}. For PromptEOL, we also choose a second part of the wording, as the PromptEOL strategy requires that the summary be in one word: {in one word, with a single word, succinctly with one word, in a unique one-word way, in a single word, in a word}.

3. Choose a separator, which include colons, commas, newlines.

4. Choose a prefix, which includes markers to indicate the first and appearance of the input.

5. Classical prompts have the form: "{instruction} {separator} {prefix} $S$".

6. Echo prompts have the form: "{instruction} {separator} {prefix0} $S$ {separator} {prefix1}$S'$".

7. PromptEOL prompts have the form: "{instruction0} {separator} {prefix} $S$ {instruction1} {separator}".

**Classical embeddings.**    For classical, we choose the prompts:

| Write a sentence I] $S$ |
|---|

| Write a prompt!
(I) $S$ |
|---|

| Write some text
PROMPT-$S$ |
|---|

| SAY A PARAGRAPH — SENTENCE 0] $S$ |
|---|

| Say a query QUERY: $S$ |
|---|

| Say a sentence!
[A] $S$ |
|---|

| COMPLETE THE PROMPT Text (1) $S$ |
|---|

| COMPLETE THE QUERY text 0) $S$ |
|---|

| Complete the query SENTENCE 0) $S$ |
|---|

| Complete the sentence:-$S$ |
|---|

| Explain a query text 0 $S$ |
|---|

| Explain a prompt — Sentence 1¿ $S$ |
|---|

EXPLAIN A SENTENCE Prompt (1) $S$

**Echo embeddings.** For echo, we choose the prompts:

Repeat The Paragraph.
query 1) $S$.
query 2) $S'$

Repeat the response.
1) $S$.
AGAIN 2) $S'$

REPEAT THE SENTENCE :: PROMPT
$S$ :: RESPONSE
S'

Rewrite the query — QUERY (A) $S$ — (B) $S'$

Rewrite the text. SENTENCE A) $S$. B) $S'$

Rewrite the response — query A] $S$ — query B] $S'$

Rephrase the sentence:@$S$:Again@S'

Rephrase The Sentence! Text ¡¿ $S$! Answer ¡¿ $S'$

REPHRASE THE QUERY Sentence a) $S$ Answer b) $S'$

Fill in the blank in the prompt: Query a) $S$: Query b) $S'$

FILL IN THE BLANK IN THE RESPONSE — Sentence A) $S$ — Sentence B) $S'$

Fill in the blank in the paragraph.
Text — $S$.
Response — $S'$

**PromptEOL embeddings.** For PromptEOL, we use the prompts:

SUMMARIZE THE QUERY.
Prompt: $S$'IN A WORD.

Summarize the sentence!
PROMPT ¡1¿ $S$'Succinctly With One Word!

SUMMARIZE THE PARAGRAPH. PROMPT (0) $S$'IN A WORD.

CATEGORIZE THE PROMPT query
$S$ With a single word

Categorize the query — prompt [1] $S$ in a word —

CATEGORIZE THE SENTENCE.
Prompt ¡1¿ $S$ IN A WORD.

Understand the sentence
@$S$ In a single word

Understand The Prompt:QUERY [0] $S$ in a single word:

UNDERSTAND THE PARAGRAPH:Text I] $S$ Succinctly with one word:

Analyze the sentence.
Sentence $S$ In A Unique One-word Way.

Analyze the response! query a¿ $S$ IN A UNIQUE ONE-WORD WAY!

Analyze The Prompt
Sentence a¿ $S$ In a unique one-word way

**Subset of MTEB for zero-shot sensitivity evaluation.** Given that we evaluate on hundreds of prompts, we choose a subset of MTEB to reflect each of the MTEB categories but with a substantially reduced computational requirement for evaluation. We evalaute on the following subset of MTEB: FiQA2018, SCIDOCS, SciFact, NFCorpus, TwitterSemEval2015, TwitterURL-Corpus, ImdbClassification, AmazonReviewsClassification, TweetSentimentExtractionClassification, MTOPDomainClassification, TwentyNewsgroupsClustering, BiorxivClusteringS2S, MedrxivClusteringS2S, StackOverflowDupQuestions, AskUbuntuDupQuestions, SciDocsRR, BIOSSES, STS12, STS13, STS14, STS15, STS16, STS17, STS22, STSBenchmark, and SICK-R. Note that this excludes the largest retrieval datasets in MTEB, on which we find PromptEOL is most likely to fail.

**Measuring the sensitivity of different embedding strategies to prompting.** We plot the sensitivity of classical, repetition, and PromptEOL to different choices of prompts for different models in Figures 7, 8, and 9. We also extend to plotting on all tested datasets individually in Figures 10, 11, and 12. We observe that PromptEOL is highly sensitive to the exact prompt used. However, neither classical nor echo were particularly sensitive. Consistently, mean token pooling outperformed last token pooling by a large factor.

## E  TOY VERIFICATION

In Section 3, we demonstrate that classical embeddings overestimate the similarity between examples which are superficially similar based on tokens that appear early in the sequence. In this section, we investigate whether this failure mode generalizes to real data. We provide qualitative and quantitative evidence that this failure mode does, in fact, generalize to real data.

### E.1  QUALITATIVE COMPARISON OF CLASSICAL AND ECHO EMBEDDINGS.

To build intuition that this applies to realistic data, we present the sentence pair from STSBenchmark, a sentence similarity task from MTEB, in which echo embeddings reduce error the most:

$x_1$: The best thing you can do is to know your stuff.  $x_2$: The best thing to do is to overcome the fussiness.

which has a ground-truth score of 0 (out of 5) similarity. The sentence pair for which echo embeddings reduces error the least is:

$y_1$: Sometime if you really want it you might need to pay an agency to get the place for you.  $y_2$: You could probably get a tour agency to do it for you but it would cost you.

which has a ground-truth similarity of 2 (out of 5). We provide more examples in the Appendix Table 11.

For these examples, the sentence pair $(x_1, x_2)$ on which echo embeddings improve error the most has exactly the property we identify as a failure mode for classical embeddings: the sentence is superficially similar for the first few tokens. On the other hand $(y_1, y_2)$ does not have this property.

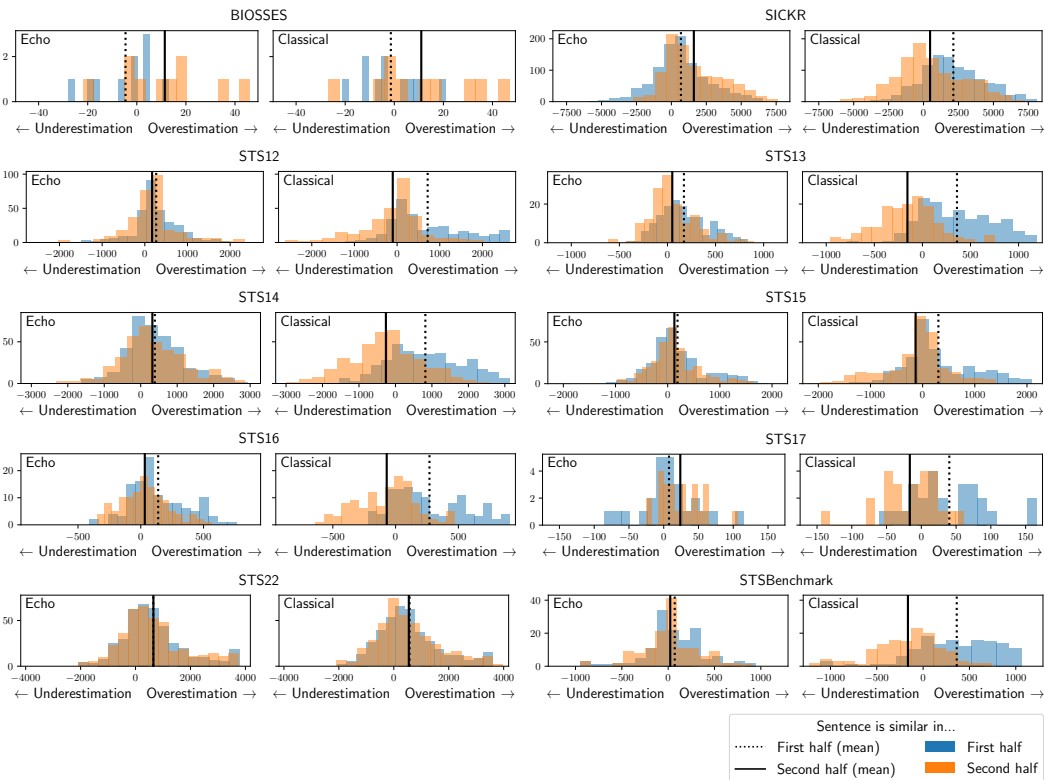

Figure 6: We plot the histogram distribution of the difference between the predicted rank and the ground truth rank of sentence pairs in STS datasets. When predicted rank is larger than the ground truth rank—when the rank difference is positive—then the embedding has overestimated the similarity of this pair. Similarly, negative values imply that the the rank is underestimated. We plot the distribution of these ranks for both classical and echo embeddings where we split the data into two groups: one in which sentences are similar in the first part of the sentence (top 10% by first-half similarity), and another in which sentences are similar in the second part of the sentence (top 10% by second-half similarity).

### E.2 Quantitative evaluation of the failure mode.

In order to test this hypothesis quantitatively, we design an experiment to identify real examples that are superficially similar in their early tokens, and examples that do not have this property. We compare the behavior of classical embeddings on these datapoints.

To conduct this experiment, we extract a set of examples from the STS datasets included in the MTEB benchmark in which the first half of the sentence is similar, and measure the degree to which the similarity is overestimated. We also select points which are similar in the second half of the sentence, and measure the degree to which similarity is overestimated. By comparing the degree to which sentences which are similar in the first half are overestimated in similarity, and the degree to which sentences which are similar in the second half are overestimated, then we can identify if classical embeddings overestimate similarity in specifically sentences which are similar in the first half. Thus, under our hypothesis, we expect that, for classical embeddings, sentences which are similar in the first half are overestimated in similarity more than sentences that are similar in their second half. On the other hand, we expect that, for echo embeddings, the degree to which similarity is over- or underestimated is independent of whether the sentences are similar in the first or second half of the sentence.

**Identifying examples based on similarity in the first/second part of the sentence.** We aim to determine which sentences are most similar in the first half of the sentence or in the second half of the sentence. For each sentence pair $x, y$, we split the sentences in half by number of words, yielding $x = [x_1, x_2]$, and $y = [y_1, y_2]$. We select sentences which are most similar in the first half by using the off-the-shelf masked-language-model-based embedding model bge-base-en-v1.5 (Xiao et al., 2023b). To select sentences that are similar in the first half, we measure the cosine similarity $\text{Sim}(x_1, y_1)$ and take the top 10% of sentence pairs $x, y$ which have the highest cosine similarity. Similarly, to select sentences which are similar in the second half, we collect the top 10% of examples by $\text{Sim}(x_2, y_2)$. We collect examples from each of the STS datasets in MTEB.

**Measuring sentence similarity estimation error.** We must determine the degree to which classical and echo embeddings overestimate similarity. The STS datasets contain sentences pairs which are ranked by similarity: the sentences which are most similar have the highest ground-truth ranking, and the least similar sentences have the lowest. We will denote the ranking of sentence pair $i$ as $r_i$. We compute an estimated ranking $\{\hat{r}_i\}$ by ranking sentence pairs by the cosine similarity between their embeddings. We can compare the error in our estimated ranking by taking the rank difference $\text{Err}_i = \hat{r}_i - r_i$. When $\text{Err}_i > 0$, we say that the $i$th sentence pair is overestimated in similarity, and similarly underestimated when $\text{Err}_i < 0$.

**Results.** We plot the the distribution over rank differences for sentences which are similar in the first half and sentences which are similar in the second half for echo and classical embeddings, from all STS datasets (Figure 6). We also highlight the means of the distributions. In accordance with our hypothesis, we observe that for classical embeddings, sentences which are similar in the first half are generally overestimated in similarity more than sentences which are similar in the second half of the sentence, suggesting that classical embeddings fail particularly on sentences that are similar in early tokens. Further, we generally observe no difference between the estimation error distributions for echo embeddings, which demonstrates that echo embeddings recover from this particular failure mode.

There are some notable counterexamples: BIOSSES does not exhibit this trend, but has few examples and thus the results may arise from noise alone. Further, STS22 exhibits identical distributions in estimation error between sentences which are similar in the first half and sentences which are similar in the second half, for both classical and echo embeddings. It is unclear why this trend fails to hold for STS22. Nonetheless, the trend holds for every other dataset, suggesting that the conceptual failure of classical embeddings that we identified in Section 3 generalizes to real data.

**Qualitative examples.** In addition, we provide qualitative examples of sentence pairs from STS-Benchmark where echo embeddings reduce error most, and where echo embeddings reduce error least, in comparison to classical embeddings. More precisely, we plot the top and bottom 7 exam-

ples ranked by $|\mathrm{Err}_i^{\mathrm{classical}}| - |\mathrm{Err}_i^{\mathrm{echo}}|$, where $\mathrm{Err}_i^{\mathrm{classical}}$ represents the rank difference of the $i$th example of classical embeddings, and $\mathrm{Err}_i^{\mathrm{echo}}$ is similar but for echo embeddings (Table 11).

### E.3 FULL ZERO-SHOT

Here we present omitted results from the main paper. First, the role of scale and base model is shown in Table 12

Table 12: *Role of scale and base model in the zero-shot setting*: We study the role of scale and base model using a different 7B parameter model: LLaMA-7B and a 1.3B parameter model: S-LLaMA-1.3B.

| Categories ⟶
# of datasets | | Clas.
12 | Clus.
11 | P. Cls.
3 | Rera.
4 | Retr.
15 | STS
10 | Su.
1 | Average
56 |
|---|---|---|---|---|---|---|---|---|---|
| Echo | 1.3B | 64.85 | 30.62 | 47.30 | 42.35 | 13.23 | 61.68 | 23.80 | **40.45** |
| Classical | 1.3B | 59.41 | 27.50 | 41.01 | 38.09 | 8.53 | 47.91 | 23.36 | 34.31 |
| PromptEOL | 1.3B | 68.86 | 24.52 | 36.52 | 43.81 | 8.34 | 56.64 | 27.36 | 37.50 |
| Echo | 7B | 70.75 | 30.68 | 65.15 | 45.72 | 18.03 | 70.19 | 29.81 | **45.84** |
| Classical | 7B | 63.54 | 31.99 | 53.78 | 42.82 | 15.07 | 55.69 | 26.21 | 40.29 |
| PromptEOL | 7B | 74.02 | 26.93 | 58.04 | 47.48 | 10.11 | 66.94 | 29.71 | 42.84 |

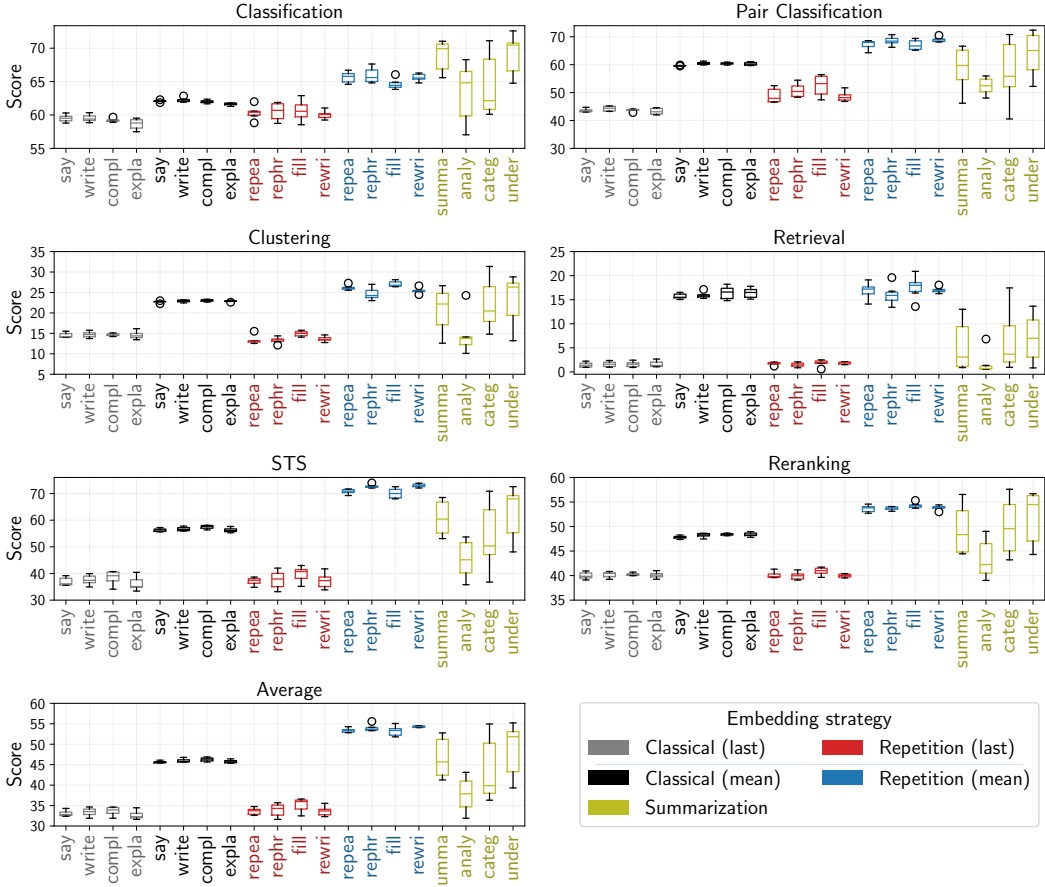

Figure 7: Variance over different prompting strategies for zero-shot Mistral-7B.

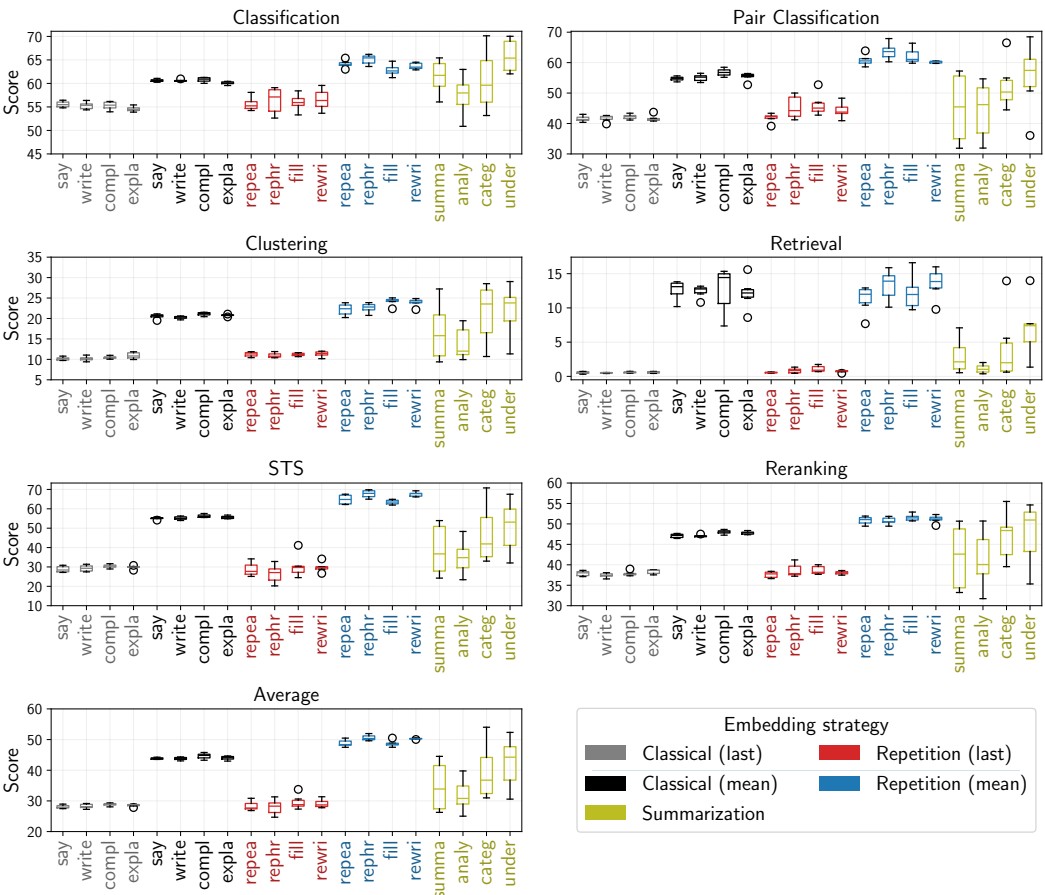

Figure 8: Variance over different prompting strategies for zero-shot LLaMa-2-7B.

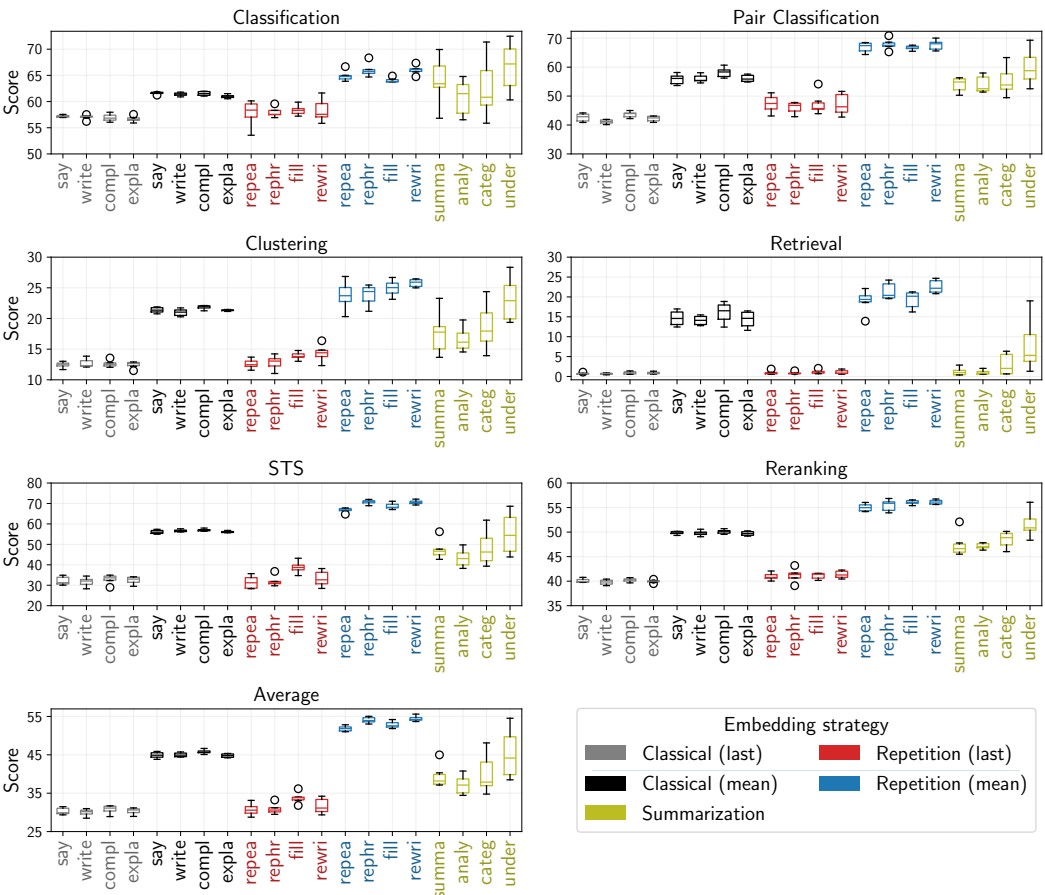

Figure 9: Variance over different prompting strategies for zero-shot LLaMa-2-13B.

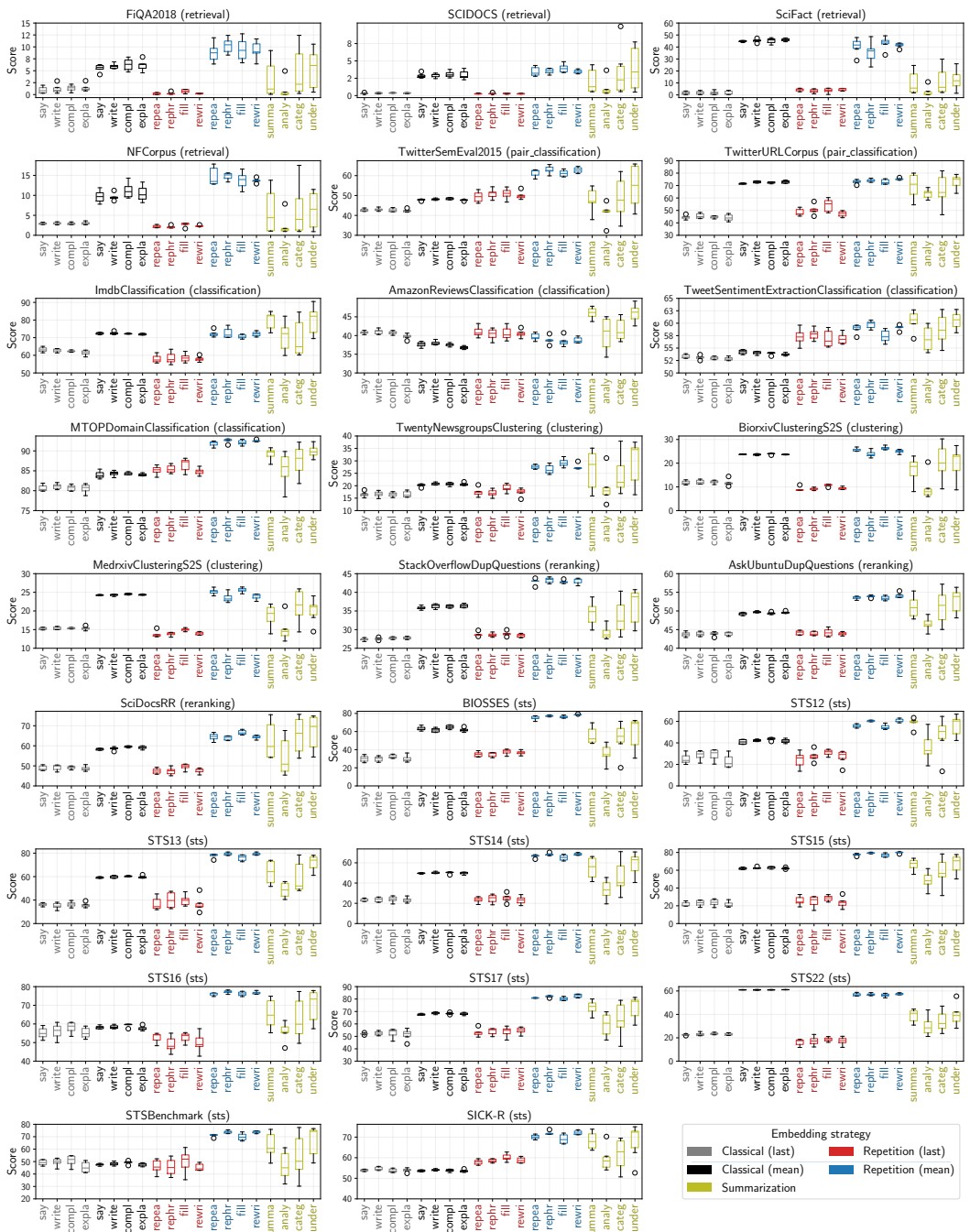

Figure 10: Variance over different prompting strategies for all evaluated datasets for zero-shot Mistral-7B.

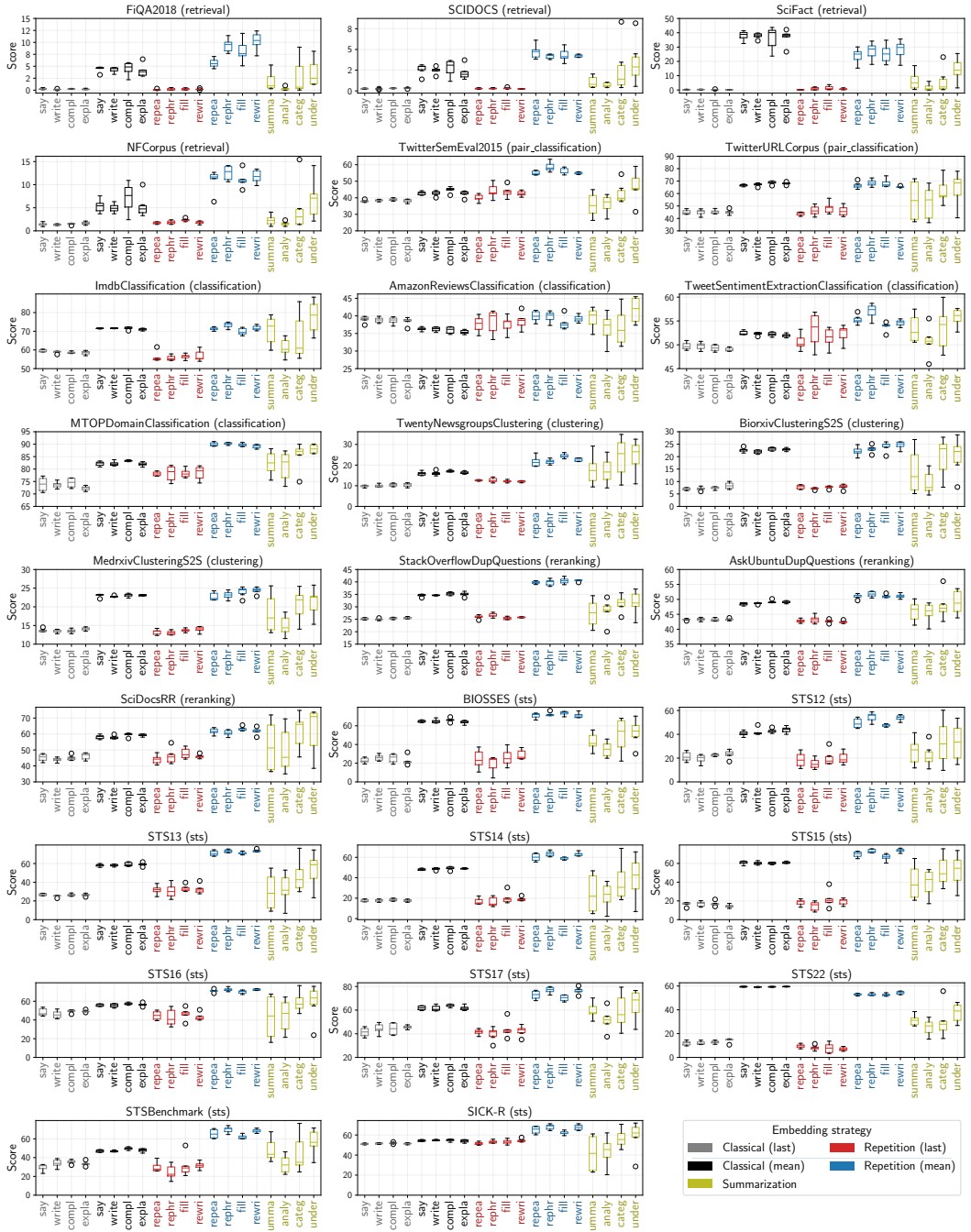

Figure 11: Variance over different prompting strategies for all evaluated datasets for zero-shot LLaMa-2-7B.

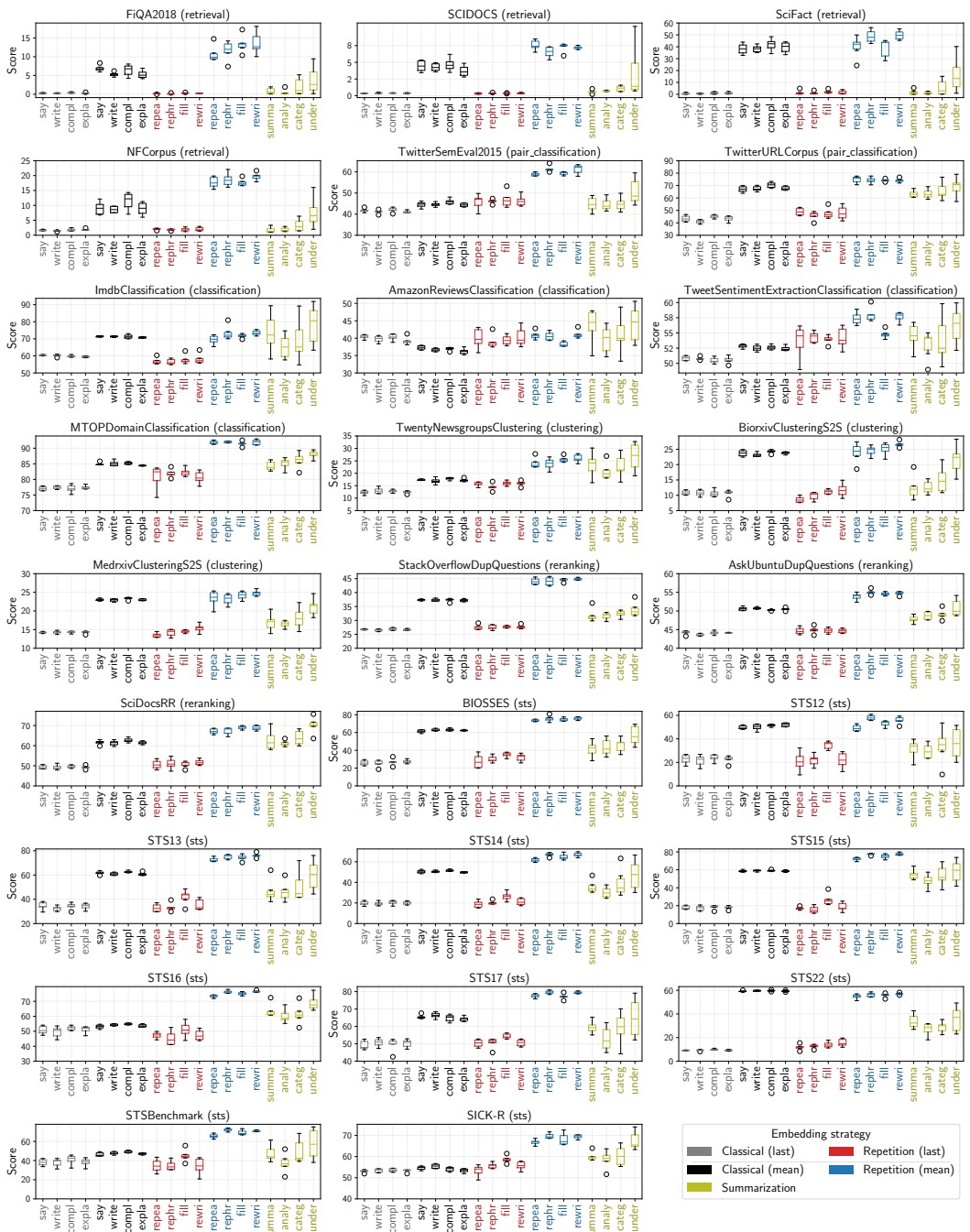

Figure 12: Variance over different prompting strategies for all evaluated datasets for zero-shot LLaMa-2-13B.

# F   ADDITIONAL FINETUNING RESULTS

Table 13: *MTEB Leaderboard:* We compare to other recent models on MTEB. For a more complete list, please visit: `https://huggingface.co/spaces/mteb/leaderboard`.

| Categories ⟶ # of datasets | Clas. 12 | Clus. 11 | P. Cls. 3 | Rera. 4 | Retr. 15 | STS 10 | Su. 1 | **Average** 56 |
|---|---|---|---|---|---|---|---|---|
| *Main fine-tuning results trained with publicly available data:* | | | | | | | | |
| Echo (ours) | 77.43 | 46.32 | **87.34** | 58.14 | **55.52** | 82.56 | 30.73 | **64.68** |
| Echo + compute matched | 77.39 | 46.27 | 87.49 | 58.08 | 55.07 | 83.18 | 30.73 | 64.66 |
| Classical | 76.57 | 45.78 | 86.37 | 56.71 | 54.87 | 82.03 | 31.02 | 63.98 |
| Classical + bidirectional attn. | 76.70 | 45.94 | **88.15** | 57.23 | 54.96 | 82.42 | 29.32 | 64.23 |
| UAE-Large-V1 (MLM) | 75.58 | 46.73 | 87.25 | 59.88 | 54.66 | 84.54 | **32.03** | 64.64 |
| multilingual-e5-large (MLM) | **77.56** | **47.10** | 86.19 | 58.58 | 52.47 | **84.78** | 30.39 | 64.41 |
| bge-large-en-v1.5 (MLM) | 75.97 | 46.08 | 87.12 | **60.03** | 54.29 | 83.11 | 31.61 | 64.23 |
| udever-bloom-7b (autoregr.) | 72.13 | 40.81 | 85.40 | 55.91 | 49.34 | 83.01 | 30.97 | 60.63 |
| sgpt-5.8b (autoregr.) | 68.13 | 40.34 | 82.00 | 56.56 | 50.25 | 78.10 | 31.46 | 58.93 |
| *Prior work trained with closed-source data (see Section 5.2):* | | | | | | | | |
| e5-mistral-7b (autoregr.) | 78.47 | 50.26 | 88.34 | 60.21 | 56.89 | 84.63 | 31.40 | 66.63 |
| GritLM-7B (autoregr.) | 79.46 | 50.61 | 87.16 | 60.49 | 57.41 | 83.35 | 30.37 | 66.76 |

In this section, we address the omitted details from the finetuning results of the main paper.

**Training Datasets.**   We follow the setup of Wang et al. (2023), and use the following datasets: ELI5 (sample ratio 0.1) (Fan et al., 2019), HotpotQA (Yang et al., 2018), FEVER (Thorne et al., 2018), MIRACL (Zhang et al., 2023b), MS-MARCO passage ranking (sample ratio 0.5) and document ranking (sample ratio 0.2) (Bajaj et al., 2018), NQ (Karpukhin et al., 2020), NLI (Gao et al., 2021b), SQuAD (Karpukhin et al., 2020), TriviaQA (Karpukhin et al., 2020), Quora Duplicate Questions (sample ratio 0.1) (DataCanary et al., 2017), Mr- TyDi (Zhang et al., 2021), DuReader (Qiu et al., 2022), and T2Ranking (sample ratio 0.5) (Xie et al., 2023). We use approximately 1.5M training examples.

**GPUs.**   Training a model takes approximately two days on four A100 GPUs. Evaluating a model on the MTEB benchmark can be completed in parallel in approximately two days with eight A100s.

**Instructions for finetuning datasets.**   We also follow the setup of Wang et al. (2023), and use the instructions in Table 15. For evaluation, we use the instructions found in Table 16.

**Models on MTEB leaderboard.**   We compare our implementation of classical and echo embeddings to state-of-the-art approaches on MTEB. Namely, we display results for UAE-Large-V1 (Li & Li, 2023a), multilingual-e5-large (Wang et al., 2024), bge-large-en-v1.5 (Xiao et al., 2023b), udever-bloom-7b (Zhang et al., 2023a), sgpt-5.8b (Muennighoff, 2022), e5-mistral-7b (concurrent work) (Wang et al., 2023).

**Pooling.**   We investigate the role of pooling for fine-tuning (Table 14). We find that last-token, with a trainable end-of-sentence token embedding, is optimal.

**Additional ablations.**   We plot additional ablations, including ablating the role of instructions during training and evaluation, as well as providing an evaluation at step 280 (out of 720 total steps), which is approximately $40\%$ of the duration of training (Table 17). We note that echo embeddings still outperform classical embeddings in this setting.

**Comparisons to the MTEB leaderboard.**   We include some recent models on the MTEB leaderboard in Table 13.

Table 14: *Role of pooling in the fine-tuning setting:* Average MTEB Score (56 datasets) for different pooling strategies. We evaluate mean and last-token pooling after fine-tuning with Mistral-7B-Instruct-v0.1.

| Categories ⟶ | Clas. | Clus. | P. Cls. | Rera. | Retr. | STS | Su. | **Average** |
|---|---|---|---|---|---|---|---|---|
| # of datasets | 12 | 11 | 3 | 4 | 15 | 10 | 1 | 56 |
| Echo + last token pooling | **77.43** | **46.32** | 87.34 | 58.14 | **55.52** | **82.56** | **30.73** | **64.68** |
| Echo + mean pooling | 77.00 | 44.94 | **87.73** | **58.30** | 55.11 | 82.52 | 29.46 | 64.22 |
| Classical + last token pooling | **76.57** | **45.78** | **86.37** | 56.71 | **54.87** | **82.03** | **31.02** | **63.98** |
| Classical + mean pooling | 76.26 | 42.68 | 86.31 | **57.58** | 53.75 | 81.53 | 30.19 | 62.96 |

| | |
|---|---|
| NLI | Given a premise, retrieve a hypothesis that is entailed by the premise |
| NLI | Retrieve semantically similar text |
| DuReader | Given a Chinese search query, retrieve web passages that answer the question |
| ELI5 | Provided a user question, retrieve the highest voted answers on Reddit ELI5 forum |
| FEVER | Given a claim, retrieve documents that support or refute the claim |
| HotpotQA | Given a multi-hop question, retrieve documents that can help answer the question |
| MIRACL | Given a question, retrieve Wikipedia passages that answer the question |
| MrTyDi | Given a question, retrieve Wikipedia passages that answer the question |
| MSMARCO Passage | Given a web search query, retrieve relevant passages that answer the query |
| MSMARCO Document | Given a web search query, retrieve relevant documents that answer the query |
| NQ | Given a question, retrieve Wikipedia passages that answer the question |
| QuoraDuplicates | Given a question, retrieve questions that are semantically equivalent to the given question |
| QuoraDuplicates | Find questions that have the same meaning as the input question |
| Squad | Retrieve Wikipedia passages that answer the question |
| T2Ranking | Given a Chinese search query, retrieve web passages that answer the question |
| TriviaQA | Retrieve Wikipedia passages that answer the question |

Table 15: Instructions for finetuning datasets.

**Performance over training time.** We plot the performance over the duration of training for a subset of MTEB tasks in Figure 13. Surprisingly, task performance *decreases* over training for many tasks.

**All results.** We plot the results for every MTEB dataset for echo embeddings, for classical embeddings, and for bidirectional embeddings in Table 18.

**Training objective.** For the training objective, we use the SimCSE loss (Gao et al., 2021b). It is defined,

$$\ell_i = -\log \frac{\exp\left(\text{Sim}\left(h_i, h_i^+\right)/\tau\right)}{\sum_{j=1}^{N} \exp\left(\text{Sim}\left(h_i, h_j^-\right)/\tau\right)}. \tag{3}$$

In this loss function, $h_i$ represents a query (or a reference sentence when the data is symmetric), $h_i^+$ represents a positive example associated with $h_i$, and $\{h_j^-\}_{j=1}^{N}$ represents the set of negatives associated with the example, including mined hard negatives.

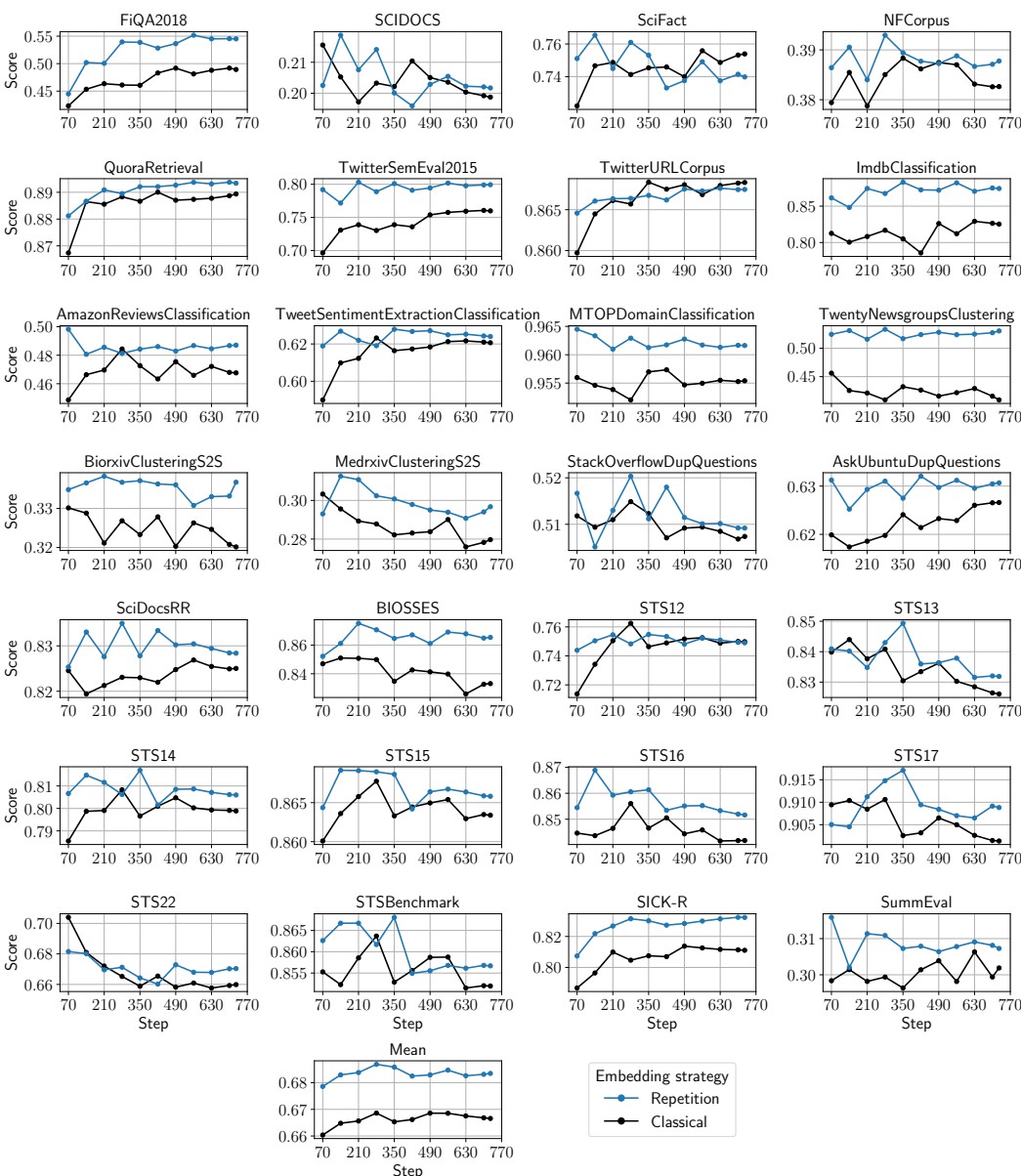

Figure 13: Performance of the evaluated MTEB datasets for finetuning over the number of finetuning steps.

| | |
|---|---|
| AmazonCounterfactualCls. | Classify a given Amazon customer review text as either counterfactual or not counterfactual |
| AmazonPolarityCls. | Classify Amazon reviews into positive or negative sentiment |
| AmazonReviewsCls. | Classify the given Amazon review into its appropriate rating category |
| Banking77Cls. | Given a online banking query, find the corresponding intents |
| EmotionCls. | Classify the emotion expressed in the given Twitter message into one of the six emotions: anger, fear, joy, love, sadness, and surprise |
| ImdbCls. | Classify the sentiment expressed in the given movie review text from the IMDB dataset |
| MassiveIntentCls. | Given a user utterance as query, find the user intents |
| MassiveScenarioCls. | Given a user utterance as query, find the user scenarios |
| MTOPDomainCls. | Classify the intent domain of the given utterance in task-oriented conversation |
| MTOPIntentCls. | Classify the intent of the given utterance in task-oriented conversation |
| ToxicConversationsCls. | Classify the given comments as either toxic or not toxic |
| TweetSentimentExtractionCls. | Classify the sentiment of a given tweet as either positive, negative, or neutral |
| ArxivClusteringP2P | Identify the main and secondary category of Arxiv papers based on the titles and abstracts |
| ArxivClusteringS2S | Identify the main and secondary category of Arxiv papers based on the titles |
| BiorxivClusteringP2P | Identify the main category of Biorxiv papers based on the titles and abstracts |
| BiorxivClusteringS2S | Identify the main category of Biorxiv papers based on the titles |
| MedrxivClusteringP2P | Identify the main category of Medrxiv papers based on the titles and abstracts |
| MedrxivClusteringS2S | Identify the main category of Medrxiv papers based on the titles |
| RedditClustering | Identify the topic or theme of Reddit posts based on the titles |
| RedditClusteringP2P | Identify the topic or theme of Reddit posts based on the titles and posts |
| StackExchangeClustering | Identify the topic or theme of StackExchange posts based on the titles |
| StackExchangeClusteringP2P | Identify the topic or theme of StackExchange posts based on the given paragraphs |
| TwentyNewsgroupsClustering | Identify the topic or theme of the given news articles |
| SprintDuplicateQuestions | Retrieve duplicate questions from Sprint forum |
| TwitterSemEval2015 | Retrieve tweets that are semantically similar to the given tweet |
| TwitterURLCorpus | Retrieve tweets that are semantically similar to the given tweet |
| AskUbuntuDupQuestions | Retrieve duplicate questions from AskUbuntu forum |
| MindSmallReranking | Retrieve relevant news articles based on user browsing history |
| SciDocsRR | Given a title of a scientific paper, retrieve the titles of other relevant papers |
| StackOverflowDupQuestions | Retrieve duplicate questions from StackOverflow forum |
| ArguAna | Given a claim, find documents that refute the claim |
| ClimateFEVER | Given a claim about climate change, retrieve documents that support or refute the claim |
| CQADupstackAndroidRetr. | Given a question, retrieve detailed question descriptions from Stackexchange that are duplicates to the given question |
| CQADupstackEnglishRetr. | Given a question, retrieve detailed question descriptions from Stackexchange that are duplicates to the given question |
| CQADupstackGamingRetr. | Given a question, retrieve detailed question descriptions from Stackexchange that are duplicates to the given question |
| CQADupstackGisRetr. | Given a question, retrieve detailed question descriptions from Stackexchange that are duplicates to the given question |
| CQADupstackMathematicaRetr. | Given a question, retrieve detailed question descriptions from Stackexchange that are duplicates to the given question |
| CQADupstackPhysicsRetr. | Given a question, retrieve detailed question descriptions from Stackexchange that are duplicates to the given question |
| CQADupstackProgrammersRetr. | Given a question, retrieve detailed question descriptions from Stackexchange that are duplicates to the given question |
| CQADupstackStatsRetr. | Given a question, retrieve detailed question descriptions from Stackexchange that are duplicates to the given question |
| CQADupstackTexRetr. | Given a question, retrieve detailed question descriptions from Stackexchange that are duplicates to the given question |
| CQADupstackUnixRetr. | Given a question, retrieve detailed question descriptions from Stackexchange that are duplicates to the given question |
| CQADupstackWebmastersRetr. | Given a question, retrieve detailed question descriptions from Stackexchange that are duplicates to the given question |
| CQADupstackWordpressRetr. | Given a question, retrieve detailed question descriptions from Stackexchange that are duplicates to the given question |
| DBPedia | Given a query, retrieve relevant entity descriptions from DBPedia |
| FEVER | Given a claim, retrieve documents that support or refute the claim |
| FiQA2018 | Given a financial question, retrieve user replies that best answer the question |
| HotpotQA | Given a multi-hop question, retrieve documents that can help answer the question |
| MSMARCO | Given a web search query, retrieve relevant passages that answer the query |
| NFCorpus | Given a question, retrieve relevant documents that best answer the question |
| NQ | Given a question, retrieve Wikipedia passages that answer the question |
| QuoraRetr. | Given a question, retrieve questions that are semantically equivalent to the given question |
| SCIDOCS | Given a scientific paper title, retrieve paper abstracts that are cited by the given paper |
| SciFact | Given a scientific claim, retrieve documents that support or refute the claim |
| Touche2020 | Given a question, retrieve detailed and persuasive arguments that answer the question |
| TRECCOVID | Given a query on COVID-19, retrieve documents that answer the query |
| BIOSSES | Retrieve semantically similar text |
| SICK-R | Retrieve semantically similar text |
| STS12 | Retrieve semantically similar text |
| STS13 | Retrieve semantically similar text |
| STS14 | Retrieve semantically similar text |
| STS15 | Retrieve semantically similar text |
| STS16 | Retrieve semantically similar text |
| STS17 | Retrieve semantically similar text |
| STS22 | Retrieve semantically similar text |
| STSBenchmark | Retrieve semantically similar text |
| SummEval | Given a news summary, retrieve other semantically similar summaries |

Table 16: MTEB instructions for evaluation of finetuned models.

Table 17: Additional ablations for finetuning. All ablations were performed using Mistral-7B as a backbone. We include ablations to compare the effect of using instructions, the effect of using the mean or last token, and the effect of using the last token at a different step.

| Categories ⟶ | Clas. | Clus. | P. Cls. | Rera. | Retr. | STS | Su. | **Avg** |
|---|---|---|---|---|---|---|---|---|
| # of datasets | 12 | 11 | 3 | 4 | 15 | 10 | 1 | 56 |
| Echo (w/ instruct., mean) | 77.00 | 44.94 | 87.73 | 58.30 | 55.11 | 82.52 | 29.46 | 64.22 |
| Echo (w/ instruct., last) | 77.43 | 46.32 | 87.34 | 58.14 | 55.52 | 82.56 | 30.73 | 64.68 |
| Classical (w/ instruct., mean) | 76.26 | 42.68 | 86.31 | 57.58 | 53.75 | 81.53 | 30.19 | 62.96 |
| Classical (w/ instruct., last) | 76.57 | 45.78 | 86.37 | 56.71 | 54.87 | 82.03 | 31.02 | 63.98 |
| Echo (w/out instruct., mean) | 75.26 | 42.93 | 86.95 | 57.05 | 55.65 | 81.40 | 30.62 | 63.28 |
| Echo (w/out instruct., last) | 75.30 | 42.94 | 86.31 | 57.31 | 54.18 | 80.92 | 31.00 | 62.80 |
| Classical (w/out instruct., mean) | 75.23 | 41.79 | 85.24 | 56.31 | 53.24 | 80.97 | 30.64 | 62.19 |
| Classical (w/out instruct., last) | 75.01 | 42.70 | 85.69 | 56.64 | 53.29 | 80.92 | 30.91 | 62.37 |
| Echo (w/ instruct., mean, step 280) | 76.84 | 45.76 | 87.72 | 59.33 | 53.55 | 82.64 | 30.33 | 64.04 |
| Echo (w/ instruct., last, step 280) | 76.41 | 46.70 | 87.17 | 59.10 | 54.84 | 82.98 | 31.09 | 64.50 |
| Classical (w/ instruct., mean, step 280) | 76.18 | 42.99 | 85.44 | 57.63 | 53.96 | 82.53 | 29.94 | 63.19 |
| Classical (w/ instruct., last, step 280) | 76.54 | 46.22 | 86.70 | 57.79 | 53.73 | 82.22 | 30.13 | 63.87 |

Table 18: Results from all MTEB datasets for finetuning with Mistral-7B.

| Dataset | Echo (last) | Echo (mean) | Classical (last) | Classical (mean) | Bidirectional (last) |
|---|---|---|---|---|---|
| AmazonCounterfactualClassification | 82.97 | 82.91 | 80.82 | 82.21 | 83.07 |
| AmazonPolarityClassification | 90.98 | 88.25 | 92.55 | 90.37 | 90.83 |
| AmazonReviewsClassification | 48.71 | 49.41 | 48.75 | 46.76 | 47.94 |
| Banking77Classification | 88.15 | 88.06 | 87.95 | 87.69 | 88.17 |
| EmotionClassification | 52.18 | 51.51 | 50.66 | 49.23 | 52.09 |
| ImdbClassification | 87.42 | 84.80 | 83.18 | 82.53 | 83.02 |
| MassiveIntentClassification | 79.67 | 79.70 | 78.60 | 79.15 | 78.93 |
| MassiveScenarioClassification | 82.82 | 82.74 | 81.71 | 81.46 | 81.80 |
| MTOPDomainClassification | 96.16 | 96.10 | 95.92 | 95.54 | 96.14 |
| MTOPIntentClassification | 85.75 | 85.87 | 85.96 | 85.86 | 85.98 |
| ToxicConversationsClassification | 71.91 | 72.21 | 71.19 | 72.21 | 71.46 |
| TweetSentimentExtractionClassification | 62.40 | 62.46 | 61.60 | 62.07 | 60.97 |
| ArxivClusteringP2P | 47.02 | 45.52 | 46.73 | 45.80 | 47.03 |
| ArxivClusteringS2S | 43.52 | 42.32 | 43.99 | 40.73 | 42.14 |
| BiorxivClusteringP2P | 35.53 | 35.24 | 36.50 | 35.42 | 36.21 |
| BiorxivClusteringS2S | 35.34 | 33.70 | 34.87 | 32.03 | 34.77 |
| MedrxivClusteringP2P | 30.27 | 29.68 | 30.67 | 29.74 | 31.06 |
| MedrxivClusteringS2S | 29.67 | 27.73 | 29.75 | 27.97 | 30.12 |
| RedditClustering | 61.77 | 59.12 | 61.17 | 54.79 | 62.50 |
| RedditClusteringP2P | 66.01 | 65.44 | 64.84 | 63.68 | 65.45 |
| StackExchangeClustering | 72.04 | 71.21 | 71.87 | 66.99 | 71.58 |
| StackExchangeClusteringP2P | 35.29 | 34.07 | 33.08 | 31.47 | 34.98 |
| TwentyNewsgroupsClustering | 53.04 | 50.29 | 50.07 | 40.91 | 49.53 |
| SprintDuplicateQuestions | 94.59 | 95.05 | 94.38 | 95.29 | 96.26 |
| TwitterSemEval2015 | 79.93 | 80.73 | 77.18 | 75.98 | 80.80 |
| TwitterURLCorpus | 87.50 | 87.40 | 87.56 | 87.67 | 87.38 |
| AskUbuntuDupQuestions | 64.13 | 64.44 | 62.24 | 63.32 | 62.65 |
| MindSmallReranking | 32.92 | 32.11 | 32.68 | 32.52 | 32.53 |
| SciDocsRR | 83.68 | 84.15 | 81.60 | 83.01 | 82.36 |
| StackOverflowDupQuestions | 51.84 | 52.51 | 50.33 | 51.48 | 51.35 |
| ArguAna | 58.52 | 56.52 | 57.22 | 51.14 | 57.27 |
| ClimateFEVER | 34.56 | 37.07 | 31.10 | 30.31 | 32.73 |
| CQADupstackRetrieval | 46.91 | 46.48 | 45.11 | 43.30 | 46.52 |
| DBPedia | 46.83 | 48.19 | 45.18 | 46.80 | 46.76 |
| FEVER | 91.22 | 91.14 | 90.30 | 90.63 | 91.66 |
| FiQA2018 | 54.51 | 54.11 | 50.31 | 48.94 | 53.06 |
| HotpotQA | 76.41 | 75.75 | 72.95 | 68.50 | 75.30 |
| MSMARCO | 43.25 | 43.11 | 42.31 | 41.49 | 43.38 |
| NFCorpus | 39.55 | 37.18 | 39.32 | 38.53 | 38.61 |
| NQ | 62.31 | 61.51 | 62.07 | 60.65 | 63.69 |
| QuoraRetrieval | 89.34 | 89.33 | 89.04 | 88.94 | 89.57 |
| SCIDOCS | 20.17 | 17.73 | 19.34 | 19.88 | 19.69 |
| SciFact | 73.99 | 73.57 | 74.22 | 75.39 | 75.83 |
| Touche2020 | 18.52 | 18.92 | 24.46 | 19.44 | 15.79 |
| TRECCOVID | 76.66 | 76.02 | 80.17 | 82.30 | 74.50 |
| BIOSSES | 86.54 | 86.78 | 85.73 | 83.31 | 85.38 |
| STS12 | 76.13 | 75.89 | 75.84 | 76.23 | 75.50 |
| STS13 | 83.19 | 82.90 | 83.41 | 82.61 | 83.44 |
| STS14 | 80.60 | 80.99 | 79.80 | 79.89 | 81.35 |
| STS15 | 87.16 | 87.16 | 86.99 | 86.68 | 87.43 |
| STS16 | 85.16 | 84.93 | 83.93 | 84.18 | 85.34 |
| STS17 | 90.88 | 90.78 | 91.12 | 90.14 | 90.99 |
| STS22 | 67.04 | 67.21 | 66.27 | 65.99 | 66.32 |
| STSBenchmark | 85.67 | 85.87 | 84.96 | 85.20 | 85.45 |
| SICK-R | 83.23 | 82.70 | 82.22 | 81.11 | 82.97 |
| SummEval | 30.73 | 29.46 | 31.02 | 30.19 | 29.32 |

