# OpenReview forum: "Repetition Improves Language Model Embeddings"
_ICLR.cc/2025/Conference — ICLR 2025 Poster_

### Official Review · Reviewer_qkAp · 2024-10-30

**Soundness:** 3
**Presentation:** 3
**Contribution:** 3
**Rating:** 6
**Confidence:** 4

**Summary:**

The paper presents a simple and effective method called "echo embeddings" that enhances text embeddings generated by autoregressive language models without finetuning. This technique involves prompting the model with the input twice and extracting embeddings from the repeated tokens. The authors demonstrate that echo embeddings significantly improve the quality of embeddings in zero-shot settings, achieving nearly the same performance as bidirectionally-converted LMs that undergo additional masked-language modeling training. Echo embeddings are also shown to be compatible with supervised fine-tuning, matching or outperforming the bidirectionally-converted LMs in direct comparisons.

**Strengths:**

- The paper introduces a novel and straightforward method to enhance autoregressive embeddings without the need for architectural changes or additional fine-tuning. This simplicity is a significant advantage, making the approach easily applicable to existing models. The results indicate that echo embeddings provide a substantial performance boost in zero-shot settings compared to classical embeddings.

- Echo embeddings can be applied in both zero-shot and fine-tuned settings, demonstrating flexibility and robustness across different tasks and benchmarks. The technique also shows consistent results across multiple models and scales. While repetition doubles the compute cost, the paper provides compute-matched results showing that echo embeddings still outperform classical embeddings even when adjusting for compute. This suggests that the method is efficient and scalable.

- The paper presents extensive experiments on a variety of datasets and tasks such as classification, clustering, reranking, retrieval, and sentence similarity. This comprehensive evaluation supports the validity and generalizability of the proposed method.

**Weaknesses:**

While the method shows strong performance across various benchmarks, it is possible that the specific tasks and datasets used in the evaluation may favor the proposed approach. Further validation on different types of data and tasks would strengthen the claim of general applicability. And also, does this technique helps some generation tasks? For example, if echoes some important information in the prompt, will it boost the performance on downstream generation tasks?

**Questions:**

- See above, performance on generation tasks.

- Can you provide more details on the impact of different prompts on the performance of echo embeddings? Have you explored variations in the prompt structure, and how sensitive are the results to these variations?

- Have you tested the echo embeddings method on a wider range of autoregressive models beyond those reported in the paper? How does the method perform on models with different architectures and training paradigms? For example, non-gpt models.

---

> ### Author Response · Authors · 2024-11-19
>
> Thanks for your review, we appreciate the feedback!
>
> >  It is possible that the specific tasks and datasets used in the evaluation may favor the proposed approach.
>
> We hope to convince you that our evaluation is very broad! We evaluate on the Massive Text Embedding Benchmark (MTEB), which is a standard benchmark to evaluate the broad quality of embeddings. It includes 56 different datasets from 7 different categories—classification, pair classification, clustering, reranking, retrieval, sentence similarity, and summarization [1]. We include a detailed breakdown of the performances across categories in Table 1&5, and by specific dataset in Appendix F. Currently, this is the gold-standard benchmark in the field for embeddings.
>
>
> > And also, does this technique helps some generation tasks? For example, if echoes some important information in the prompt, will it boost the performance on downstream generation tasks?
>
> Great question! Our primary focus is on constructing high quality embeddings, and thus we do not evaluate the generation capabilities. However, commonly, methods to convert large language models into embedding models require modifying the architecture by converting causal to bidirectional attention and also additional training. In our setting, we require no changes to the architecture, which means that a base language model, with strong generation capabilities, could be used to construct high quality embeddings while also retaining its ability to perform well on generation tasks. We believe echo embeddings are thus an important step toward unified machine learning models, but investigating generation tasks in the context of echo embeddings is a promising avenue for future work.
>
>
> > Can you provide more details on the impact of different prompts on the performance of echo embeddings? Have you explored variations in the prompt structure, and how sensitive are the results to these variations?
>
> Absolutely! We find that as long as we adopt the echo embeddings structure from Section 3.2—that is, repeat the input twice and instruct the model to perform a repetition-like task—the model is fairly robust to changes in the exact wording or structure. Based on [2], we evaluate echo embeddings with randomly generated prompts which vary in the instruction (we vary the language in the instruction, including using: “repeat”, “rephrase”, “fill in the blank”, and “rewrite”, along with minor variations in the phrasing), and also the exact formatting details, such as the separators between the instructions and each input (e.g., newlines, colons, other characters) and capitalization. We find that echo embeddings have similar performance for most prompts—the variance is low (Figure 3). See Appendix D for further details.
>
> We perform the same analysis for our baselines—classical embeddings, and PromptEOL. Classical embeddings, while also robust to changes in the exact prompting format, have much lower performance than echo embeddings, and PromptEOL is highly sensitive to the exact structure of the prompt.
>
>
> > Have you tested the echo embeddings method on a wider range of autoregressive models beyond those reported in the paper? How does the method perform on models with different architectures and training paradigms?
>
> Our primary focus, at the moment, is on autoregressive transformer-based models, as transformer-based models are currently the models that perform best on benchmarks on general NLP tasks (e.g., [3]). We do evaluate over two different transformer-based architecture classes (Mistral and LLaMA). Evaluating on, e.g., Mamba-style models would be an interesting avenue for future work!
>
>
> [1] Muennighoff, Niklas, et al. "MTEB: Massive text embedding benchmark." arXiv preprint arXiv:2210.07316 (2022).
>
> [2] Sclar, Melanie, et al. "Quantifying Language Models' Sensitivity to Spurious Features in Prompt Design or: How I learned to start worrying about prompt formatting." arXiv preprint arXiv:2310.11324 (2023).
>
> [3] Dubey, Abhimanyu, et al. "The llama 3 herd of models." arXiv preprint arXiv:2407.21783 (2024).

---

> > ### Comment · Reviewer_qkAp · 2024-11-26
> >
> > Thank you for your rebuttal! I'm expecting to see more experimental results. Currently I'll keep my score as 6.

---

### Official Review · Reviewer_HXgu · 2024-11-03

**Soundness:** 3
**Presentation:** 3
**Contribution:** 3
**Rating:** 6
**Confidence:** 4

**Summary:**

Causal attention in autoregressive models limits the use of LLMs in text embedding tasks because tokens in a sequence cannot reference subsequent tokens. This paper introduces an "echo embedding" method to transform autoregressive language models into high-quality text embedding models without altering the causal attention architecture or requiring fine-tuning. By repeating the input sequence twice and extracting embeddings from the second occurrence, the tokens can attend to the full input from the first occurrence. Experimental results demonstrate that the proposed method can nearly match the performance of bi-directional and fine-tuned models using zero-shot embeddings.

**Strengths:**

1. The proposed method is simple by repeating the input sentence twice to get the text embeddings.

2. The toy example design is interesting.

3. The results of zero-shot settings is impressive.

**Weaknesses:**

1. The motivation regarding causal attention seems questionable. Although LLM2Vec utilizes causal attention, it still performs exceptionally well in extracting text embeddings.

2. After fine-tuning the model, the performance gap between "echo embedding" and other models is minor. However, "echo embedding" requires the input sentence to be repeated twice, increasing computational costs. This limitation confines the proposed method to zero-shot settings only.

3. At least one illustrative example should be included in the main article, rather than relegating all examples to the appendix.

4. Typo: line 122, "encode can" -> "can encode"
line 283: "embedding 4" -> "embedding in Table 4"

**Questions:**

1. I am curious about the influence of number of repetition times, which is not important though.

---

> ### Author Response · Authors · 2024-11-19
>
> Thanks for your review, we appreciate the feedback!
>
> > The motivation regarding causal attention seems questionable. Although LLM2Vec utilizes causal attention, it still performs exceptionally well in extracting text embeddings.
>
> Good question. We want to clear up a possible misconception here: LLM2Vec does not utilize causal attention. LLM2Vec is based on Mistral, but modified to have bidirectional attention (see Figure 1 left in their paper [1]).
>
> This is what makes echo embeddings exciting—the requirement to modify the architecture to be bidirectional, which is common among recent autoregressive large embedding models [1, 2], is unnecessary—echo embeddings can be effective even without changing their architecture
>
>
> > After fine-tuning the model, the performance gap between "echo embedding" and other models is minor. However, "echo embedding" requires the input sentence to be repeated twice, increasing computational costs.
>
> We find that echo embeddings can outperform classical embeddings even in an entirely compute-matched setup. In particular, we can achieve a compute-matched setup by only encoding the first half of each input. The improvement of embedding quality is enough, with echo embeddings, that even when only encoding the first half of the input, performance is still improved over classical embeddings (Table 5, “compute matched” row). In addition, even during training, we can train for fewer steps to match the compute cost of classical embeddings. Even when combining both of these tricks so that the training- and inference-time compute are matched, echo embeddings still outperforms classical embeddings.
>
>
> > I am curious about the influence of number of repetition times
>
> Great question (also asked by Reviewer gzEy)! We ran this experiment, and found no improvement past the first repetition; results are below. Note: For this rebuttal, to save on compute, we evaluate on a 37-dataset subset of MTEB (typically 56 datasets). For this reason, the numbers themselves are not comparable to any other numbers in the paper.
>
> | # of repetitions |	Average (37 datasets) |
> | - | - |
> | Classical (0x) |	46.04 |
> | Echo (1x)  |      53.06 |
> | Echo+ (2x) | 	52.97 |
> | Echo++ (3x) |	52.43 |
> | Echo+++ (4x) |	51.75 |
>
>
> > At least one illustrative example should be included in the main article, rather than relegating all examples to the appendix.
> > Typo: line 122, "encode can" -> "can encode" line 283: "embedding 4" -> "embedding in Table 4"
>
> Thanks for pointing these out, we will make these changes.
>
>
> [1] BehnamGhader, Parishad, et al. "Llm2vec: Large language models are secretly powerful text encoders." arXiv preprint arXiv:2404.05961 (2024).
>
> [2] Muennighoff, Niklas, et al. "Generative representational instruction tuning." arXiv preprint arXiv:2402.09906 (2024).

---

> > ### Comment · Reviewer_HXgu · 2024-11-20
> >
> > Thanks for the feedback. I mistakenly misused the examples of LLM2Vec. What I intended to ask is why a classical LLM, fine-tuned with causal attention, performs exceptionally well at extracting text embeddings. (63.98 with causal attention v.s. 64.23 with bidirectional attention, the gap is minor)

---

> > > ### Author Response · Authors · 2024-11-21
> > >
> > > Thanks for clarifying your original query! To make sure we’re on the same page: our main motivation for echo embeddings arises from the fact that prior state-of-the-art autoregressive-based embedding models do use bidirectional attention which requires some fine-tuning [1, 2]. A big benefit of echo embeddings is that it allows us to use bidirectional information in a zero-shot manner with no finetuning.
> > >
> > > However, you raise an interesting point that the gap between causal and bidirectional attention is quite small *after* fine-tuning on a contrastive loss for good embeddings. From a representational perspective, with last-token pooling, the last token can attend to every other token and causal embeddings can encode all useful bidirectional information in the last token. Nonetheless, even if encoding bidirectional information is possible, it isn’t necessarily the case that they *will* learn to encode the “proper” bidirectional transformation for good embeddings—for example, classical embeddings don’t do this “properly” zero-shot (see Section 3 last token pooling). In fact, this is true when using bidirectional attention, too—even though the attention explicitly enables bidirectional communication between token embeddings, there is no guarantee that the model will learn to encode the information “properly”.
> > >
> > > With enough data when fine-tuning, both causal embeddings and bidirectional embeddings can learn to encode bidirectional information in the last-token embeddings to some extent. The large-scale finetuning (1.5M examples) in our results might explain the good performance of both methods.
> > >
> > > Echo embeddings, on the other hand, show that prompting is sufficient to ensure the model “properly” processes bidirectional information. As a result, we continue to see a gap between classical and echo embeddings even after fine-tuning.
> > >
> > > As an alternative perspective, one could look at the zero-shot results as a proxy for how well these methods encode bidirectional information. The gap between echo embeddings and classical embeddings are significant and clear in the zero-shot setting, and persists (admittedly to a smaller extent) after fine-tuning.
> > >
> > > [1] BehnamGhader, Parishad, et al. "Llm2vec: Large language models are secretly powerful text encoders." arXiv preprint arXiv:2404.05961 (2024).
> > >
> > > [2] Muennighoff, Niklas, et al. "Generative representational instruction tuning." arXiv preprint arXiv:2402.09906 (2024).

---

> > > > ### Comment · Reviewer_HXgu · 2024-11-26
> > > >
> > > > I have reviewed the authors’ feedback and acknowledge their comments. However, I believe the impressive performance of the proposed method is limited to the zero-shot setting, which prevents me from assigning a score of 8. Therefore, I prefer to maintain my score of 6.

---

### Official Review · Reviewer_PgPq · 2024-11-04

**Soundness:** 2
**Presentation:** 3
**Contribution:** 3
**Rating:** 6
**Confidence:** 4

**Summary:**

This paper proposes a simple strategy to extract high-quality sentence representation from causal LMs, i.e., repeating the input and extracting embeddings from the repeated tokens. They argue that their method requires neither bi-directional attention nor supervised fine-tuning. They first conduct experiments on a toy dataset, where they can control the information type of part of the inputs. Then, they clearly show the limitation of classical embeddings (mean pooling or last token) and the advantage of their method. Experiments on the MTEB dataset show that their method has advantages over classical approaches and recent alternatives. However, I hold some concerns with the experiment setting, see weakness parts. Overall, I find the paper's idea interesting, but the practical aspects need further discussion.

**Strengths:**

1. I appreciate the toy experiment, which clearly supports their claim about the limitation of classical embeddings and the advantages of echo embeddings.

2. The results on the MTEB dataset show clear improvements over classical embedding extraction settings, achieving comparable results with LLM2Vec, which needs backbone changes and unsupervised finetuning.

3. The method itself is very simple and insightful, requiring no changes to the backbone.

**Weaknesses:**

1. The setting of the most relevant baseline, promptEOL, does not seem to exactly align with that in the original paper. The results of PromptEOL appear significantly different from those reported in the original paper. In the original study, PromptEOL achieved an average score of 72.10 across seven STS tasks using the OPT-6.7B model. However, in your paper, PromptEOL only obtains an average of 67.14 on ten STS tasks. I didn't expect such a big performance discrepancy. Is this because of the three additional STS tasks in the evaluation?

2. I found the prompt used in this paper differs from that in the original paper. How about the performance of promptEOL that uses exactly the same prompt?


3. I found the "halve input" setting weird to me. For a prompt like "rewrite S, rewritten S'", does it mean halving both S and S' to extract embedding? If so, I will read the results as the "nature"/"issue" of the problem itself, rather than an advantage of echo embedding. E.g., how about also halving the input for PromptEOL?

**Questions:**

See weaknesses

---

> ### Author Response · Authors · 2024-11-19
>
> Thanks for your review, we appreciate the feedback! To respond to your main concern: We believe we have a fair and sound comparison to PromptEOL—to verify we exactly reproduce the results of their paper below. Thanks for the great questions, in addition to our responses we will edit the paper to address everything.
>
>
> > The results of PromptEOL appear significantly different from those reported in the original paper.
>
> We agree that the difference between our evaluation and [1] (PromptEOL) is strange, and so we went ahead and triple-checked that our evaluation is correct. There are two primary reasons for the difference:
> As you mentioned, we evaluate on three additional STS datasets that are part of MTEB;
> We use Mistral-7B-Instruct-v0.1, which performs slightly worse than OPT-6.7B. This is consistent with the PromptEOL paper, where they also find, counterintuitively, that larger models sometimes have worse performance on the STS tasks.
>
> Upon closer inspection of the code from [1], we noticed that they perform a data preprocessing step to add punctuation to each example, and to replace double quotes with single quotes. To achieve identical results (“our reproduction + preprocessing” below), we include this preprocessing step. Surprisingly, we find that this preprocessing slightly harms performance.
>
> First, we plot a comparison between the original paper, and our reproduction:
>
> |                | STS12 | STS13 | STS14 | STS15 | STS16 | STS-B | SICK-R | *Mean* |
> |----------------|-------|-------|-------|-------|-------|--------------|------------------|----------|
> | **OPT-6.7B + PromptEOL (as reported in [1])**    | 60.91 | 80.05 | 67.65 | 75.49 | 80.11 | 72.91        | 67.57            | **72.10** |
> | **OPT-6.7B + PromptEOL (our reproduction + preprocessing)**    | 60.94 | 80.02 | 67.62 | 75.49 | 80.11 | 73.01        | 67.70            | **72.13** |
> | **OPT-6.7B + PromptEOL (our reproduction without preprocessing)**    | 59.54 | 79.59 | 68.54 | 75.51 | 79.48 | 74.77 | 70.26 | **72.52** |
>
>
> Now, comparing OPT to Mistral:
>
> |                | STS12 | STS13 | STS14 | STS15 | STS16 | STS-B | SICK-R | *Mean* |
> |----------------|-------|-------|-------|-------|-------|--------------|------------------|----------|
> | **OPT-6.7B + PromptEOL (our code, same as above)**    | 59.54 | 79.59 | 68.54 | 75.51 | 79.48 | 74.77 | 70.26 | **72.52** |
> | **Mistral-7B-Inst-v0.1 + PromptEOL (our code)**           | 65.23 | 77.47 | 66.82 | 75.46 | 71.20 | 71.87        | 68.33            | **70.91** |
>
>
>
> Here are the results from the other three tasks:
>
> |           | BIOSSES | STS17 | STS22 | *Mean (over all)* |
> |----------|-------|-------|--------|---------|
> | **OPT-6.7B + PromptEOL (our code)**   | 74.03 | 75.62 | 25.60 | **68.29** |
> | **Mistral-7B-Inst-v0.1 + PromptEOL (our code)** | 57.11 | 79.06 | 38.86 | **67.14** |
>
>
> > I found the prompt used in this paper differs from that in the original paper.
>
> Another good point of clarification, thank you for pointing this out. We use the exact prompt provided by the PromptEOL paper for the results in Table 1 (please see above for proof of replication), but we forgot to include this detail in our paper. We will add it in the revision.
>
> In addition to replicating their results, we also investigated the prompt sensitivity of the PromptEOL method (Figure 3). Since the original paper did not provide other variants of their prompt, we included our own systematically generated variations of the original prompt (Appendix D) in order to measure the variance in performance over the exact prompt.
>
>
> >  For a prompt like "rewrite S, rewritten S'", does it mean halving both S and S' to extract embedding?
>
> Yes, in this situation we halve both S and S’. Perhaps surprisingly, we find that the benefits in embedding quality from echo embeddings still outweigh the drawback we incur from halving the inputs on the MTEB benchmark, which consists of a broad set of evaluation datasets. In cases where halving the number of encoded tokens harms performance, echo embeddings would require additional compute, but this additional compute improves the quality of embeddings beyond the classical methods!
>
>
> > How about also halving the input for PromptEOL?
>
> This is a good point. We find that there are (small) compute budgets where classical embeddings outperform echo embeddings after limiting the number of tokens seen by the model (Figure 4) and we suspect the same would be true for PromptEOL. If compute is available, however, echo embeddings outperforms PromptEOL by ~5% (Table 1). This means that in order to achieve strong performance in this setting, echo embeddings (and the additional compute) may be necessary.
>
>
> [1] Jiang, Ting, et al. "Scaling sentence embeddings with large language models." arXiv preprint arXiv:2307.16645 (2023).

---

> > ### Author Response · Authors · 2024-12-01
> >
> > Thanks again for your review! Please let us know if we have addressed your concerns, and you have any additional questions.

---

### Official Review · Reviewer_gzEy · 2024-11-04

**Soundness:** 4
**Presentation:** 4
**Contribution:** 3
**Rating:** 6
**Confidence:** 3

**Summary:**

The paper proposes a very easy method to improve the language model embedding. They introduce echo embedding by repeating the input and extracting embedding from the repeated tokens. The experiment show that the echo embedding improve over classical LM embeddings by over 5% in zero-shot settings. Echo embeddings are also compatible with supervised fine-tuning, even with an identical compute during training and inference.

**Strengths:**

1. The echo embedding method is both easy and effective. While previous studies have demonstrated that repetition is beneficial for reasoning tasks and recurrent language models, this paper shows that it is also effective for causal language model embedding.
2. The paper is clearly written and easy to understand.
3. The use of a simple synthetic dataset to analyze why causal attention might inhibit embeddings from reliably capturing information across the entire context is interesting.

**Weaknesses:**

The echo embedding method will inevitably double the input length. Although experiments show that reducing the input length and training steps by half still yields good results, this approach may not be suitable in cases where important information is located in the latter half of the input context. For example, the S2 (Early redundant; late discriminatory) cases described in Section 3.1 of the paper. Additionally, because self-attention has a computational complexity of O(n^2) with respect to input length, training for only half as many steps may not necessarily equate to the training cost of the original baseline.

**Questions:**

Have you tried tripling or increasing the input length even further? I'm curious to know if this would result in any additional improvements.

---

> ### Author Response · Authors · 2024-11-19
>
> Thanks for your review, we appreciate the feedback!
>
> > The echo embedding method will inevitably double the input length. Although experiments show that reducing the input length and training steps by half still yields good results, this approach may not be suitable in cases where important information is located in the latter half of the input context.
>
> As you point out, there are cases where observing the entire sequence is helpful, such as S2 data (early redundant; late discriminatory). However, we find that classical embeddings can perform poorly on S2 data, too, especially when using mean-pooling (Section 3). Since mean-token is pooling is essential in the zero-shot setting (Table 4), echo embeddings may be necessary to obtain strong performance, anyways.
>
> > Because self-attention has a computational complexity of O(n^2) with respect to input length, training for only half as many steps may not necessarily equate to the training cost of the original baseline.
>
> Good point! We also report echo embeddings performance after ~⅓ of an epoch of training, and find that it still outperforms classical embeddings (Table 17, step 280). Alternatively to training for fewer steps, we can also reduce train-time compute by halving the input length, as we do to match inference-time compute. For completeness, we will add a comparison to the final manuscript (the compute requirement makes it infeasible to train and evaluate additional models during the rebuttal period).
>
>
> > Have you tried tripling or increasing the input length even further?
>
> Great question (Also asked by Reviewer HXgu)! We ran this experiment, and found no improvement past the first repetition; results are below. Note: For this rebuttal, to save on compute, we evaluate on a 37-dataset subset of MTEB (typically 56 datasets). For this reason, the numbers themselves are not comparable to any other numbers in the paper.
>
> | # of repetitions |	Average (37 datasets) |
> | - | - |
> | Classical (0x) |	46.04 |
> | Echo (1x)  |	           53.06 |
> | Echo+ (2x) | 	52.97 |
> | Echo++ (3x) |	52.43 |
> | Echo+++ (4x) |	51.75 |

---

### Meta-Review · Area_Chair_AqpL · 2024-12-23

**Metareview:**

The paper proposes “echo embedding,” a method to improve language model embeddings by repeating the input and extracting embeddings from the repeated tokens. It aims to address the limitations of causal attention in autoregressive models for text embedding tasks without architectural changes or fine-tuning.  All reviewers vote for accptance and  there is an overall agreement that the method is simple, effective in zero-shot settings, and the experiments are relatively comprehensive. However, each reviewer also raises valid concerns that need to be addressed, e.g., input length and computational complexity, which are well discussed during the rebuttal. Overall, this is a good contribution to the conference.

**Additional Comments On Reviewer Discussion:**

The concerns of reviewers come from several aspects: 1) more input length leads to inefficiency; 2) discrepancy in baseline results: 3) task and dataset bias. The authors show that even when reducing the input length and training steps, echo embeddings can still outperform classical embeddings.  The authors thoroughly explain the reasons for the performance discrepancy with PromptEOL, including dataset differences, model differences, and preprocessing steps.

---

### Decision · Program_Chairs · 2025-01-22

Accept (Poster)